# Synaptic and intrinsic membrane defects disrupt early neural network dynamics in Down syndrome

Saad B. Hannan [1,2] ✉, Ivan Alić [3,4], Aoife Murray [3], Joonhong Kwon[5], Martin Mortensen [1], Hyo Jung Kang[5], Ante Plećaš [4], Pollyanna A. Goh[3], Niamh L. O'Brien[3], Richard Naud [6], Dean Nižetić [3] ✉ & Trevor G. Smart [1] ✉

Down syndrome, caused by trisomy 21, affects around six million people worldwide and features learning, memory and language deficits. However, the mechanisms underlying trisomy 21 neurophenotypes involving human cortical circuitry are unknown. By characterising developing neural network dynamics and single cell excitability profiles, from synaptic and voltage-dependent ion channel behaviour using an isogenic induced pluripotent stem cell-derived neuronal model, we show that trisomy 21 impairs the activity and development of cortical circuitry. This is caused by deficient glutamatergic synaptic connectivity and by aberrant intrinsic membrane properties involving $K^+$ and $Na^+$ channels culminating in spike firing defects that weaken neural network activity and disrupt the synchrony of developing neurons. We also identify transiently activated A-type $K^+$ channels, specifically Kv4.3 channels, as a key orchestrator for Down syndrome during neurodevelopment. Overall, these excitability changes will significantly contribute towards the aberrant neurophenotypes observed later on in life.

Down syndrome is the most prevalent genetic cause of intellectual disability caused by trisomy of human chromosome 21 (hCh21)[1,2]. Multiple organ systems are affected by the presence of an extra copy of hCh21, but the most debilitating impact of trisomy evident in early life is reflected by the neurophenotype including: neurodevelopmental, psychiatric, neurological and neurodegenerative conditions, which are typically severe and life-changing[3]. Therefore, identifying the mechanisms that underlie these nervous system phenotypes has been a priority area for Down syndrome research. An increase in gene dosage has long been postulated to cause Down syndrome phenotypes, but in regard to neuronal excitability, there is a paucity of ion channel and neurotransmitter receptor genes located on hCh21, e.g. *KCNE1-2, TRPM2, KCNJ6, GRIK1, FNAR1, IFNAR2, IFNGR2,* and *IL10RB*[4–8]

suggesting that genome wide effects of hCh21 genes are important for establishing the mechanistic basis for Down syndrome.

To date, perturbations to several brain receptors and ion channel signalling pathways have been implicated in Down syndrome, the most prominent to date being the GABAergic signalling pathway[9–12]. Reduced cell numbers and depleted synaptic density, and altered neural innervation patterns all feature prominently[13–17], but despite this, the development of early human neural networks in Down syndrome, and how cellular and synaptic excitability is affected remains mostly unknown. Here, by using isogenic human cortical neurons derived from induced pluripotent stem cells (iPSCs)[18], we investigate neural network activity and functional profiles of single neurons. These cellular models allow critical insights into early developing cortical

[1]Department of Neuroscience, Physiology and Pharmacology, University College London, London, UK. [2]Department of Molecular and Cellular Biology, Harvard University; 52 Oxford Street Cambridge, Cambridge, MA, USA. [3]Blizard Institute, Barts and The London School of Medicine and Dentistry, Queen Mary University of London, London, UK. [4]Department of Anatomy, Histology and Embryology, Faculty of Veterinary Medicine, University of Zagreb, Zagreb, Croatia. [5]Department of Life Science, Chung-Ang University; 84 Heukseok-ro, Dongjak-gu, Seoul, Republic of Korea. [6]Department of Cellular and Molecular Medicine, University of Ottawa; 451 Smyth Rd, Ottawa, ON, Canada. ✉e-mail: saadhannan@fas.harvard.edu; d.nizetic@qmul.ac.uk; t.smart@ucl.ac.uk

networks[19–21] that are impossible to study in vivo. They are beneficial since they are human in origin and contain the same complement of genes with or without increased dosage of those on hCh21[18,22]. Our study reveals that early Down syndrome developing networks are characterised by widespread changes to excitatory (glutamatergic) synaptic connectivity and aberrant intrinsic excitability involving Na[+]- and K[+]-channels. Moreover, by deploying a combination of K[+] channel pharmacology, transcriptomics and mathematical modelling, we postulate that $K_v4.3$ channels could be key mediators of the network and excitability deficits that feature so prominently in trisomy 21.

## Results

### Developmental anomalies in neural network dynamics resulting from trisomy 21

The longitudinal developmental activity profiles of human Down syndrome neurons were probed by studying spike firing properties of two sets of isogenic cell lines derived from a Down syndrome donor mosaic

for trisomy 21, which are either euploid controls (C3 and C9) or aneuploid trisomy 21 (C5 and C13) neurons that reproduce and underpin clinical Down syndrome phenotypes[18,22,23]. To establish the activity profiles of these cell lines, extracellular electrical activity from 2D cultures was recorded using microelectrode arrays (MEAs; Fig. 1A, B) populated with neurons at 4 – 14 weeks in vitro. MEA was preferred over imaging approaches as it is a direct measure of neuronal activity rather than a proxy, and also because early-stage iPSC-derived neurons and immature neurons display developmentally regulated slow $Ca^{2+}$ oscillations[24] which can make the use of $Ca^{2+}$ to gauge action potential firing unreliable. Thus, the use of MEA recordings ensured that only action potential-dependent changes to excitability were analysed.

To validate genetic dispositions, interphase fluorescence in situ hybridisation (FISH) confirmed the integrity of the third hCh21 copy in neural stem cells (NSCs) derived from iPSCs (Figure S1A, B). These NSCs expressed cell markers SOX2, nestin and PAX6 (Figure S1C), and were differentiated into action potential-firing mature neurons

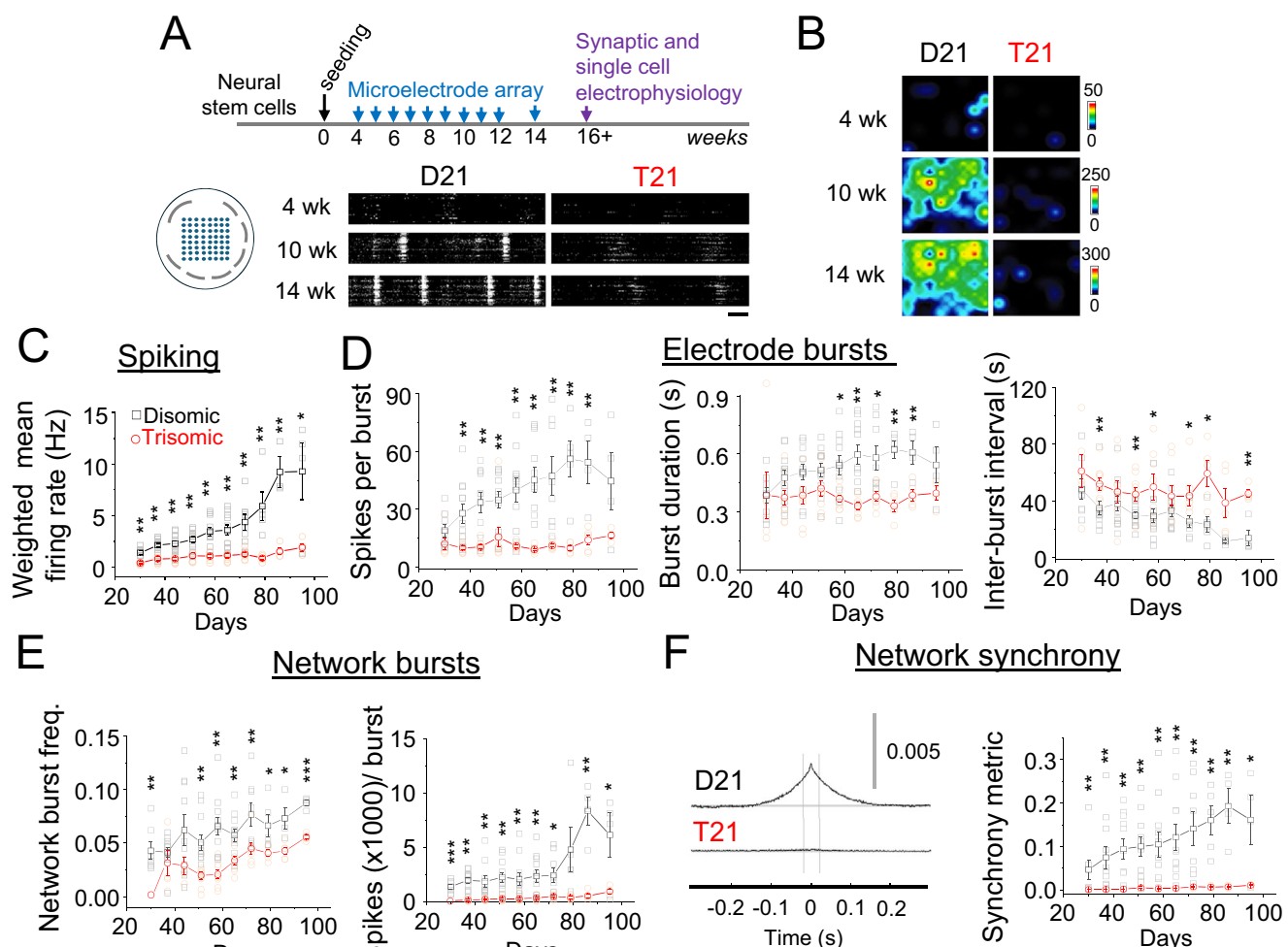

**Fig. 1 | Early developmental trisomy 21 neurons are deficient in spiking and network synchrony. A** Timeline of electrophysiological characterisation after neural stem cell seeding for terminal differentiation, along with position of 8 ×8 recording (300 μm separation) and ground electrodes (gray). Raster plots showing spikes (white) and bursting of disomic (D21) and trisomic (T21) cells at 4, 10 and 14 weeks (wks in vitro). Scale bar 5 s. **B** Heat maps of spiking detected by the recording electrodes show greater activity of disomic cells across neural networks. Heat map legends depict spike rates (spikes / second). **C** Weighted mean spiking rate of disomic and trisomic networks. **D** Single electrode bursts including spikes per burst, burst duration and inter-burst intervals of disomic and trisomic networks. **E** Network burst parameters including network burst frequency and spikes per burst of disomic and trisomic networks. **F** *Left*, Network synchrony depicted by normalised cross correlation between network bursts at 14-weeks for disomic and trisomic wells. *Right*, synchrony metric showing average area under the normalised cross-correlation curves of disomic and trisomic cells. The results here are from five different differentiations of two different lines per genotype and at least four wells per time point. $n = 4$–11 wells. Error bars in this and preceding figures, unless otherwise stated, depict standard error of means (SEM). *$P < 0.05$, **$P < 0.01$, ***$P < 0.001$, two-tailed unpaired t-test or Mann-Whitney test. All quantitative values included in this study can be found in Data S1.

expressing the pan-neuronal markers TUBB3 and MAP2 (Figure S1D–G; Data S1). These neurons were predominantly (> 93%) excitatory (Figure S1H), ascertained by the expression of VGLUT1. They exhibited no overt functional inter-clonal variations in electrophysiological properties between euploid (C3 and C9) or isogenic trisomy 21 (C5 and C13; Figure S2) and therefore we pooled the results according to their genotype. This allowed the comparison of the impact of trisomy on neural circuitry to be assessed during development.

Throughout the course of our developmental timeframe, compared to disomic counterparts, trisomic cells were deficient in action potential spiking. This was manifest by a markedly reduced spiking rate in addition to impaired bursting properties, assessed from single electrode bursts including spikes per burst, burst duration, inter-burst interval and burst frequency, all recorded using MEA (Fig. 1C, D; and Figure S3). Consequently, network bursting, defined as synchronous bursting at a minimum of 20-35% of electrodes, was reduced in number, frequency and synchrony for trisomic neurons. Moreover, network bursts were characterised by fewer spikes in the trisomic cells (Fig. 1E, F; Figure S3) implicating impaired spike firing at the single-cell level as a potential mechanism for the difference between trisomic and disomic cells. The presence of cells in direct contact with the electrode arrays was monitored throughout so we could discount an absence of cells due to cell death or mechanical shearing as an explanation for spiking and network activity deficits (Figure S3).

Network bursts and spike synchrony of disomic and trisomic neurons were blocked by the $Na^+$ channel inhibitor tetrodotoxin (0.5 $\mu$M[25]) and inhibited by the AMPA receptor and NMDA receptor antagonists, CNQX (10 $\mu$M)[25] and APV (25 $\mu$M[26]; Figure S3G–J) respectively, confirming that intrinsic and synaptic mechanisms were integral for network dynamics. These results suggest that synaptic and cellular anomalies affecting spiking could together underlie the aberrant developmental neural network dynamics due to trisomy 21.

## Defects in synaptic transmission due to trisomy 21

A wide range of synaptic deficits characterise murine models of Down syndrome including changes to GABAergic neurotransmission and synaptic plasticity[10]. However, the impact of Down syndrome on excitatory neurotransmission, particularly in the context of human neurons, has received little attention. Our iPSC differentiation yielded a majority of excitatory neurons (> 93%) and consistently these cultures were near-devoid of inhibitory interneuron activity exemplified by the absence of GABAergic synaptic activity in 88-98% of cells despite cell surface expression of functional $GABA_A$ receptors (Figure S4). We detected GABAergic postsynaptic currents in only one of eight differentiations that were blocked by the antagonist bicuculline[27] (Figure S4). Since inhibitory synaptic activity could arise due to altered cellular differentiation trajectories, for consistency, to avoid any aberrant differentiation artefacts, we excluded this batch of iPSCs from all analyses in this study.

We characterised glutamatergic neurotransmission for iPSC-derived neurons using voltage clamp recordings after they reached the plateau phase of neural network development at 4–6 months. Disomic and trisomic cells were similar in size, deduced by voltage clamp capacitance discharge curves (with trending towards smaller trisomic cells; Figure S4E). Furthermore, these cells produced functional AMPA and NMDA receptor-mediated whole-cell responses (Figure S4). Interestingly, a larger proportion of trisomic neurons (20%; Fig. 2A) did not receive any synaptic inputs compared to disomic neurons (4%) in standard saline solution. Synaptic inputs onto these neurons were exclusively excitatory characterised by their fast rise times and decay kinetics and by charge transfer[28,29] (Figure S4G). Definitively, the synaptic currents were abolished by the glutamate receptor blockers, CNQX and APV (Fig. 2B). Importantly, the frequency of trisomic excitatory postsynaptic currents (EPSCs) was only 25% of that recorded in disomic cells without any change to EPSC amplitude

or kinetics due to unchanged AMPA receptor expression or clustering (Fig. 2B–E; and Figure S4). These results suggest that during early development trisomic neurons exhibit a severe deficiency in glutamatergic synaptic connectivity without notable changes to cell surface AMPA receptor numbers.

To explore the paucity of synaptic currents in disomic and trisomic cells, we increased the excitability of our neurons by removing $Mg^{2+}$ from the bathing solution (0 $Mg^{2+}$) to relieve inhibition of NMDA receptors[30]. As expected, in the absence of $Mg^{2+}$, neurons showed greater synaptic activity but 10% of trisomic cells (compared to 0% disomic neurons) did not exhibit synaptic currents despite undergoing 0 $Mg^{2+}$-induced membrane bursting (Fig. 2F) thereby identifying a proportion of cells that fail to integrate into cortical networks during development. Here, the frequency of EPSCs for synaptically connected trisomic cells was approximately 10% of those for disomic cells. The EPSC amplitude, kinetics and AMPA and NMDA current densities were unchanged (Fig. 2G–J; and Figure S4) confirming a likely deficit in synaptic connectivity involving glutamatergic synapses. The absence of NMDA-dependent synaptic transmission, in some neurons, could reflect silent synapses and will impact on synaptic plasticity with consequences for learning and memory.

This reduced connectivity facet was confirmed by immunolabelling for the presynaptic marker synapsin-1[31] and excitatory postsynaptic density protein PSD95[32] which had reduced fluorescence in trisomic compared to disomic neurons (Fig. 2K, and Figure S5). These results suggest that impaired integration of glutamatergic cells into developing networks and reduced synaptic connectivity underlies the neural network defects in trisomy 21.

Analysis of single cell bursting in 0 $Mg^{2+}$ also revealed deficits in network dynamics consistent with our MEA results. Using this approach, all disomic neurons studied exhibited bursting, a feature that is prevalent in mature neocortical neurons[33] but approximately a third of trisomic cells did not exhibit membrane current oscillations (Fig. 3A). These oscillations correspond to cellular depolarisation following population action potential spiking[34] and can be abolished by blocking AMPA receptors, NMDA receptors or restoring $Mg^{2+}$ to the saline (Fig. 3B; and Figure S6A–C) suggesting that network bursting is dependent on glutamatergic synaptic activity. Trisomic network bursts recorded in postsynaptic voltage clamp were less frequent (by 33%) compared to disomic bursts and had nearly 66% reduction in current amplitudes and 70% decrease in charge transfer (Fig. 3C–F) similar to our MEA results (Fig. 1).

Together these results suggest that trisomy 21 results in a reduction in glutamatergic synapses and synaptic activity, and during development a proportion of cells fail to connect into neural networks. This reduced connectivity is also manifest by the impaired bursting properties of trisomic neurons.

## Intrinsic membrane and $Na^+$ channel defects in human Down syndrome cells

Glutamatergic activity is important for network development, but since intrinsic membrane properties can determine the overall state of excitability, strength and synchronicity of neural networks, the excitability and spike firing properties of single neurons was explored using current clamp recordings. Trisomic neurons displayed a trend towards more depolarised resting membrane potentials even though the input resistance, membrane time constant and capacitance were unchanged (Fig. 4A; and Figure S6D). Although a difference of 6 mV in median resting potentials between disomic and trisomic cells could be relevant physiologically, injecting constant current steps to elicit spikes, revealed that the rheobase of the two cell types was unchanged (Fig. 4B). However, injecting current steps as increments of the rheobase revealed spike firing differences. Trisomic cells fired fewer action potentials compared disomic cells (Fig. 4C, D) in the absence of changes to the spike firing threshold (Fig. 4E).

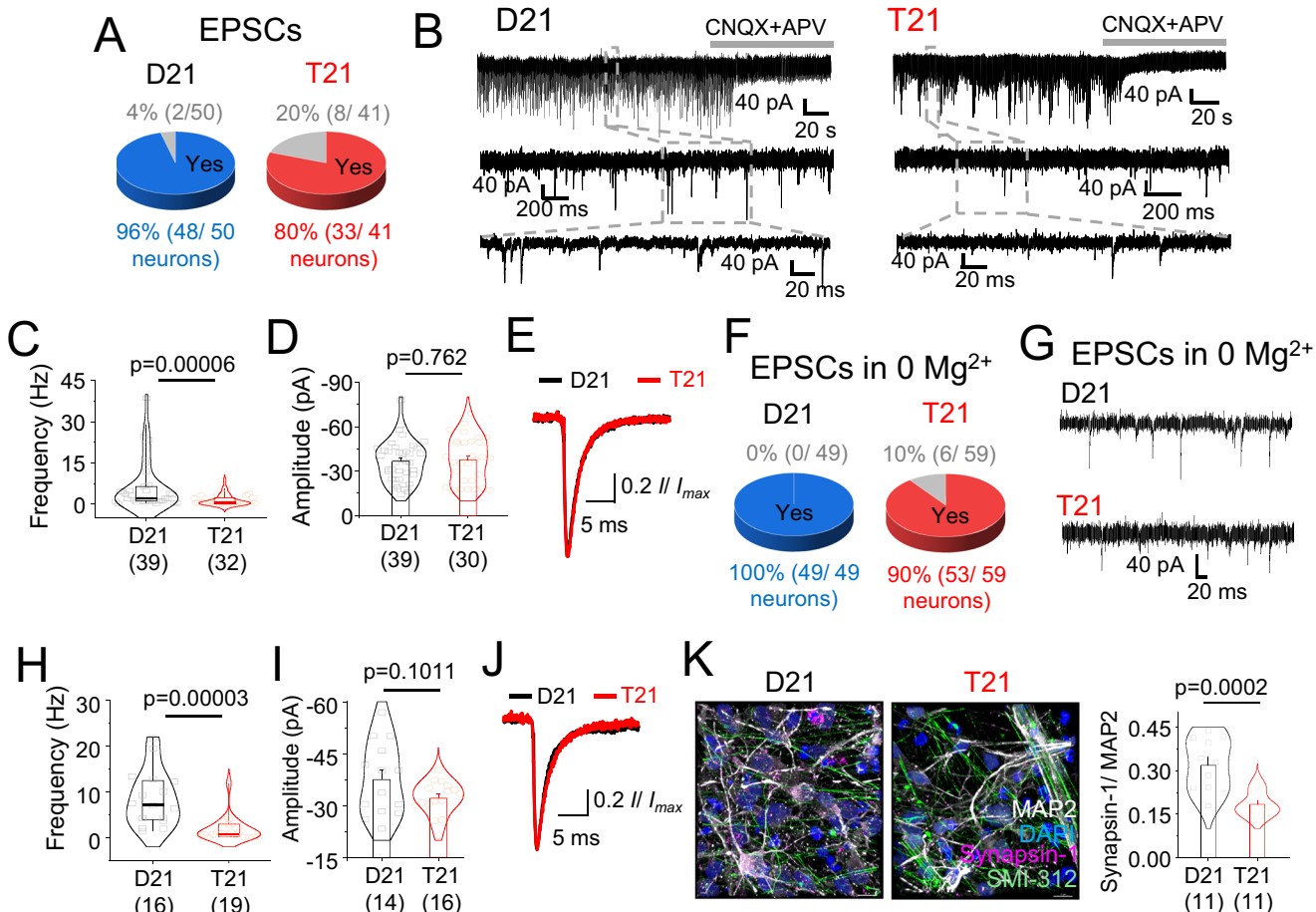

**Fig. 2 | Synaptic and developmental wiring defects due to trisomy 21. A** Pie charts showing proportion of disomic (D21) and trisomic (T21) cells exhibiting excitatory postsynaptic currents (EPSCs). **B** EPSCs were blocked by CNQX (10 μM) and APV (25 μM) and had fast kinetics with no residual GABAergic currents in the presence of glutamatergic blockers. **C–E** Reduced frequency (**C**) but not amplitude (**D**) and kinetics (**E**) of EPSCs in T21 cells in normal saline. **F** Pie charts showing that all D21 cells received EPSCs in $Mg^{2+}$-free saline but 10% of T21 cells did not. **G** Representative EPSC recordings of D21 and T21 cells outside bursts in 0 $Mg^{2+}$. **H–J**

Reduced frequency (**H**) but not amplitude (**I**) or kinetics (**J**) of EPSCs of T21 cells in 0 $Mg^{2+}$ saline. **K** Confocal images and localisation of presynaptic terminal marker synapsin-1 with pan-axonal neurofilament marker SMI-312 and dendritic marker MAP2 counterstained with nuclei marker DAPI in D21 and T21 cells. Scale bars 10 μm. $n$ = 11–59; number of cells/ 3D image stacks. N numbers of 3D image stacks have been depicted in brackets along with $p$ values on graphs; two-tailed unpaired t-test or Mann-Whitney test. Box plots in C, H show median, 25–75% interquartile range and 2-95% whiskers. Bar charts in D, I and K depict means +/- SEM.

By injecting 5, 10 or 20 pA constant current steps, disomic cells tolerated larger current injections and fired more spikes with increased current injections (Fig. 4F). The latency of spiking for trisomic cells was lower at the rheobase (Fig. 4G) and spike jitter was greater (Fig. 4H). These results suggest that impaired spike firing could underlie altered bursting in trisomic networks with increased spike jitter compromising network synchrony.

Analysing action potential waveforms by studying the first action potential generated at the rheobase also revealed changes to multiple parameters. Compared to disomic spikes, trisomic action potential amplitudes were lower but the spike area was greater due to more prolonged activation rise times and decays exemplified by the time to decay to 50% of the maximum spike amplitude ($T_{50}$; Fig. 4I–M). These results imply the involvement of $Na^+$ and $K^+$ channel defects that could underlie impaired spike waveforms associated with trisomy 21.

$Na^+$ channel activity was examined by directly depolarising neurons using a voltage step in the absence and presence of tetrodotoxin. The resulting tetrodotoxin-sensitive current density was comparable between disomic and trisomic cells suggesting unchanged gating of $Na^+$ channels (Fig. 5A). Similarly, $Na^+$ current-voltage (I-V) relationships did not reveal changes to reversal potentials or current densities (Fig. 5B). However, both the fast and slow inactivation profiles of trisomic $Na^+$ channels were shifted to more hyperpolarised potentials

(Fig. 5C, D) suggesting that reduced excitability of trisomic $Na^+$ channels could underlie defects in spiking.

These results identify $Na^+$ channel and spike firing defects in addition to changes in membrane properties that could underlie neural network deficits that typify trisomy 21 neurons.

## Impaired fast-transient $K^+$ channel conductance due to $K_V4.3$ downregulation in human Down syndrome cells

Prolonged action potential waveforms, reduced latency and increased jitter are all indicative of voltage-gated $K^+$ channel defects and therefore we characterised $K^+$ channel I-V relationships in disomic and trisomic neurons (Fig. 6A, B). The $K^+$ current density of trisomic neurons was lower compared to disomic cells suggesting that $K^+$ channels are indeed reduced in these Down syndrome cells. We next focused on transient A-type $K^+$ currents as these have fast onsets, they are activated at high voltages, they bring neurons rapidly back towards their resting membrane potentials and because they can influence action potential half-width, latency and jitter[35–37]. Measuring the difference of currents in response to a voltage step to 120 mV from a holding potential of −110 mV where A-type channels can be activated, or from −10 mV where these channels are inactivated, revealed transient $K^+$ currents in both disomic and trisomic cells (Fig. 6C, D). Importantly, the A-current density of trisomic cells was lower, without changes to

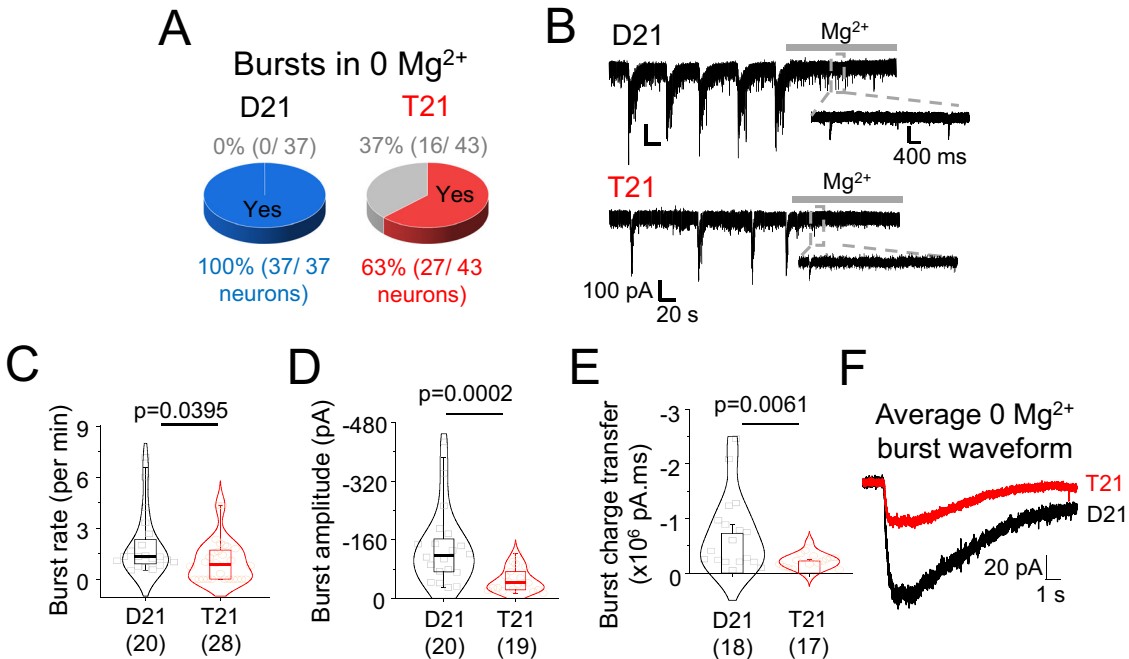

**Fig. 3 | Altered bursting of trisomy 21 neurons. A** Pie charts showing that all disomic (D21) cells had bursts in Mg$^{2+}$-free saline but around a third of trisomic (T21) cells did not. **B** Representative traces of membrane bursts of D21 and T21 cells in Mg$^{2+}$-free saline. Introduction of Mg$^{2+}$ saline abolishes bursts but not EPSCs (insets). **C**–**F** Lower bursting rate (**C**), burst amplitude (**D**), burst charge transfer (**E**) and altered burst kinetics (**F**) of T21 neurons compared to D21 cells. $n = 17$–43 cells have been depicted in brackets along with $p$ values on graphs; two-tailed unpaired t-test or Mann-Whitney test. Box plots in (**C**, **D**) depict median, 25–75% interquartile range and 2–95% whiskers. Bar chart E depicts means +/- SEM.

voltage dependence of activation and inactivation (Figure S6E−G), compared to disomic cells suggesting that this key current is depressed in early developing trisomic neurons.

A-currents are mediated by the following gene products: K$_v$1.4 (gene - *KCNA4*), K$_v$3.3-3.4 (*KCNC3-4*) and K$_v$4.1-4.3 (*KCND1-3*), which have variable sensitivities to the K$^+$ channel blockers tetra-ethylammonium (TEA; 1 mM) and 4-aminopyridine (4-AP; 3 mM) such that K$_v$3.3-3.4 and K$_v$1.4 homomers are blocked by TEA, whereas K$_v$4.1-4.3 are insensitive to TEA[38]. Application of TEA revealed that disomic cells are minimally sensitive to TEA (15% inhibition), whilst the sensitivity to this antagonist is increased by more than two-fold for trisomic cells. The level of A-current inhibition is markedly increased for both disomic and trisomic cells by adding 4-AP diminishing the differential sensitivity observed with TEA alone (Fig. 6E). Together with a lower A-current density of trisomic cells (Fig. 6D), these data suggest that a TEA less-sensitive A-current which is expressed in disomic cells, is less apparent in trisomic cells, possibly involving a change in relative levels of K$^+$ channel expression.

Using an unbiased approach, we analysed RNA expression metadata in an unrelated post-mortem human brain tissue dataset[15,39] and confirmed genome wide alterations affecting K$^+$ channel, Na$^+$ channel and glutamatergic synapse genes (Figure S7A, B) throughout life (Table S1, and Data S2−6). Importantly, applying a P < 0.01 cut-off revealed that TEA-insensitive *KCND3* is down-regulated in dorso-lateral prefrontal cortex (DFC) but not in the cerebellar cortex (Fig. 7A; and Figure S6C) at early developmental stages (Fig. 7B). By comparison, the other TEA-insensitive A-current genes are unaffected at these stages in the DFC (Figure S8A). Interestingly, the DFC expression of other A-current genes such as the TEA-sensitive *KCNC4* is also low at early stages whereas *KCND2* and *KCNC3* expression is normal in early development but decreases later on in life (Fig. 7C−E). Notably, there are no apparent changes to the expression of hyperpolarisation-activated cyclic nucleotide–gated (HCN) channel genes through life or evident in their functional activity in iPSC

neurons (Figures S8B-C). These channels can also alter oscillations in neural networks[40]. Thus, in summary, our results identify a new ion channel component, K$_v$4.3 (encoded by *KCND3*), as a mediator for cell-intrinsic neurodevelopmental defects in the developing Down syndrome brain.

We corroborated these findings in our iPSC-derived neurons and consistent with findings in early-developmental post-mortem tissues, the levels of *KCND3* mRNA, probed by qPCR using three sets of primers (Fig. 7F, and Data S1), were reduced in trisomic neurons compared to their disomic counterparts. Moreover, the level of Kv4.3 protein expression was also reduced in trisomic neurons, probed using Western blots (Fig. 7G, and S9A) and immunofluorescence (Fig. 7H, and S9B) analysis, whereas the levels of TEA-sensitive Kv4.2 (*KCND2* gene product) was unchanged (Figure S9C) as would be predicted during early development ascertained by post-mortem analysis.

Finally, we simulated a recurrently connected network of excitatory rate-based units that generate spontaneous intrinsic bursts according to a Poisson process. The interconnections developed according to standard Hebbian plasticity. We modelled trisomy 21 by reducing intrinsic bursting since this is thought to be controlled by levels of Kv4.3[41,42]. The rate of network bursts and the connection probability was assessed as a function of simulated days. This simple model reproduced the impaired transition to a highly connected network producing synchronous bursts (Fig. 7I). This suggests that small perturbations to Kv4.3 can have large effects on developing neural networks.

## Discussion

By differentiating isogenic iPSCs to derive human neurons from an individual with mosaic Down syndrome, we have identified early functional deficits in cellular and neural networks that occur due to hCh21 trisomy. We have recorded the activity of hundreds of neurons from seven iPSC differentiations to ensure variability is limited – a feature that has been reported in iPSC derived neurons[43]. Our findings

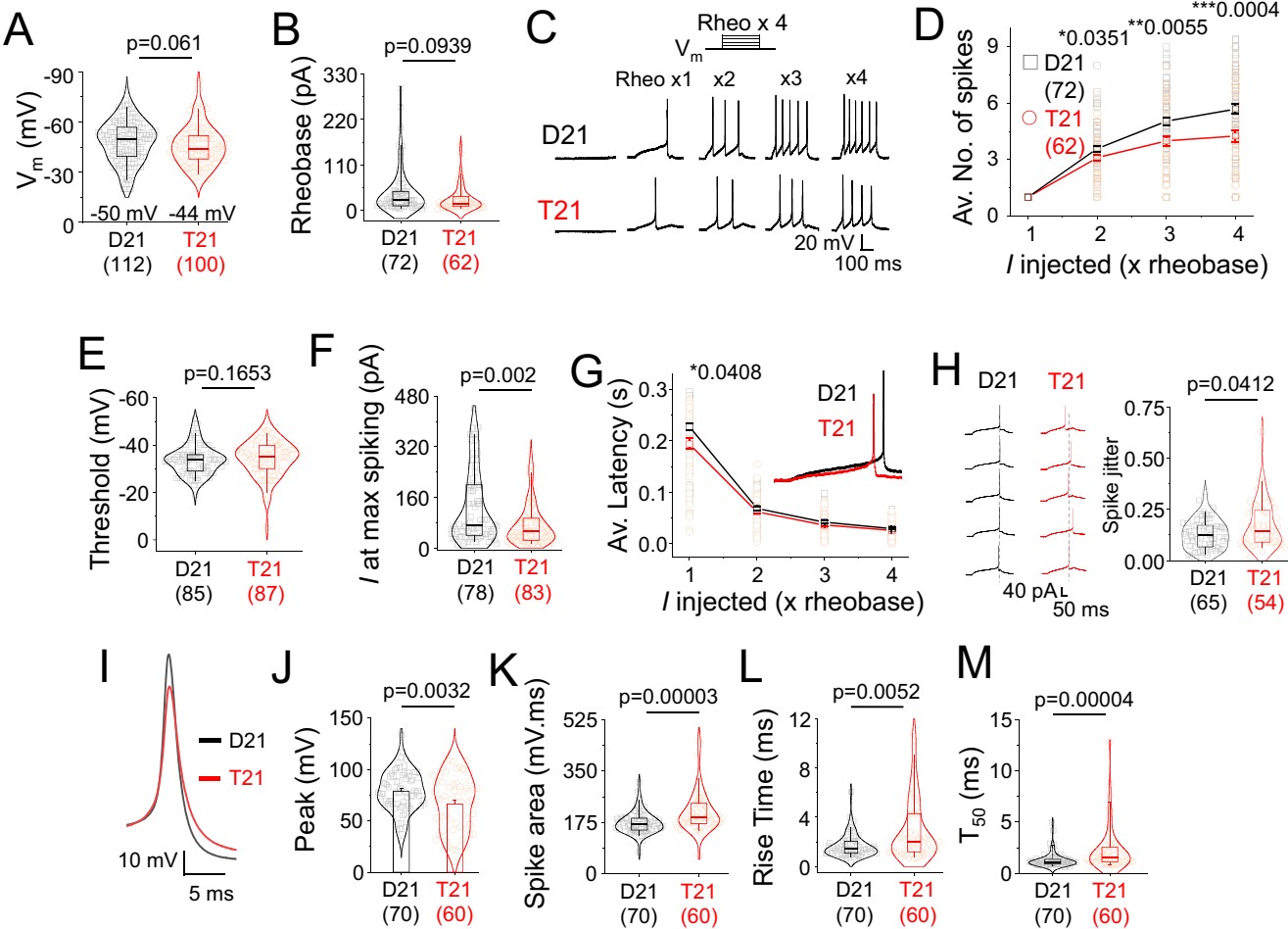

**Fig. 4 | Spike firing anomalies due to trisomy 21. A** Resting membrane potential of disomic (D21) and trisomic (T21) neurons. **B** Rheobase of D21 and T21 neurons. **C** Spike outputs elicited by injecting depolarising steps of currents as increments of rheobase (rheo). **D** Input-output relationships of spikes and current injection. **E** Threshold potential at which neurons fire action potentials at rheobase. **F** Current at which maximum spikes are elicited in a step protocol of fixed current increments. **G** Average latency of first spike for the rheobase step current injection protocol. **H** Example of spike jitter traces and average spike jitter at rheobase for D21 ($n = 65$) and T21 ($n = 54$) cells. **I–M** Average action potential waveform (**I**), peak potential (**J**), area (**K**), rise time (**L**) and $T_{50}$ (**M**) of D21 and T21 spikes at rheobase. $n = 54$-112 cells. *$P < 0.05$, **$P < 0.01$, ***$P < 0.001$, two-tailed unpaired t-test or Mann-Whitney test. Here, box plots show the median, 25–75% interquartile range and 2–95% whisker and the bar chart depicts means +/- SEM.

provide unique insights into how the early development of Down syndrome afflicts human neurons and their excitability.

## Excitability of early Down syndrome networks

Using differentiated iPSCs that are suited to studying early human development[19] we demonstrate that early Down syndrome neurons are characterised by a plethora of changes to excitability and network dynamics at crucial stages of development which likely underpin long-term changes in brain function. While these cellular models can be limited due to their in vitro nature, iPSCs do recapitulate several key developmental neurobiology facets, and additionally, our central findings are consistent with and validated by results from postmortem human brain tissue studies and by mathematical modelling. We found that Down syndrome neurons are characterised by aberrant glutamatergic synaptic connectivity; the exclusion of neurons from developing neural networks; and reduced burst firing even in states favouring elevated excitability by using zero $Mg^{2+}$ bathing solution. These perturbations dampen the overall connectivity, excitability and plasticity of developing cortical networks. The state of synchronous bursting reported in these early differentiated neurons in vitro is physiologically relevant as it is reminiscent of early neonatal activity states of in vivo networks[44]. At the single cell level, a shift in $Na^+$ channel inactivation was observed that will further reduce cell excitability.

Moreover, reduced fast-transient $K^+$ channel function, most likely due to at least a down-regulation of $K_v4.3$ channels will reduce spike latency affecting spike bursting according to computational modelling[41,42]. Trisomic neurons are also compromised by reduced glutamate receptor based synaptic activity and innervation that are required to generate and sustain high frequency network activity in early developmental circuits.

## Involvement of non-hCh21 $Na^+$/ $K^+$ channel and glutamate synapse genes

hCh21 gene dose-related alterations to neural development and function have been linked to various Down syndrome phenotypes[2,14,45–47]. Despite this, alterations to the expression of genes that are integral for neural function including, neurotransmitter receptor and ion channel genes, along with their functional activity, have not been characterised particularly in the context of human tissues. This is important as there are only a handful of these ion channel genes present in hCh21 suggesting the involvement of genes elsewhere in the human genome that leads to alterations of neural function in early Down syndrome networks. Using metadata RNA expression analysis of $Na^+$ channel, $K^+$ channel and glutamate synapse genes of post-mortem human DFC and cerebellar tissues we found that many non hCh21 genes are differentially regulated in Down syndrome across various

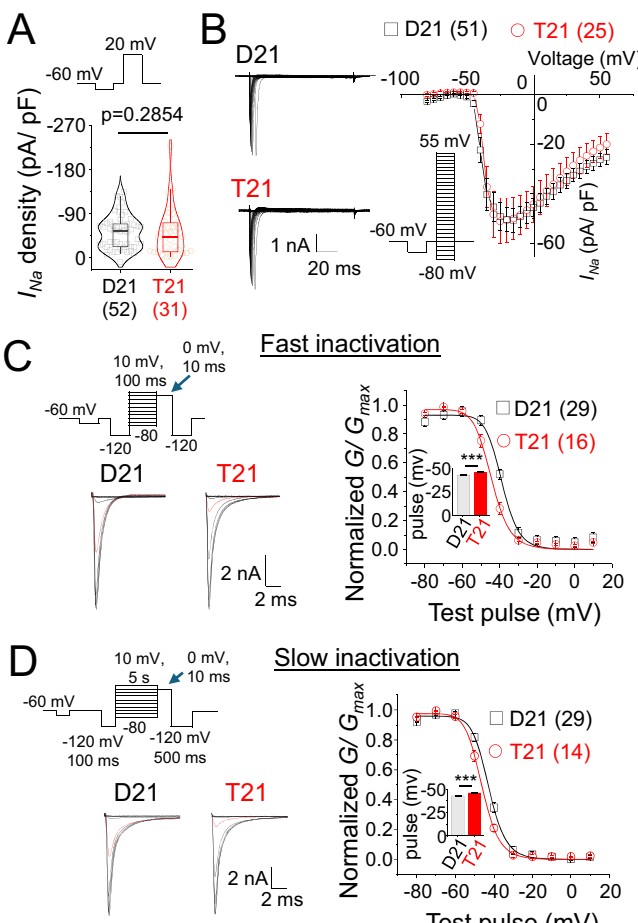

**Fig. 5 | Voltage-gated Na⁺ channel inactivation anomalies due to trisomy 21.**
**A** TTX-sensitive Na⁺ current density of D21 and T21 cells evoked by a single depo-
larising step. **B** TTX-sensitive Na⁺ currents across a range of voltages along with the
voltage-step protocol and I-V curve of current density. **C** Step protocol for fast
(100 ms) inactivation and example traces of Na⁺ channel recovery from fast inac-
tivation immediately following test pre-pulse (blue arrow in step protocol). The
trace with −40 mV pre-pulse is shown in red. Graph shows the voltage dependence
of fast inactivation of D21 and T21 cells. Inset, Half-maximal fast inactivation
potential. **D** Step protocol for slow (5 s) inactivation and example traces of Na⁺
channel recovery from slow inactivation following pre-pulse (blue arrow in step
protocol). The trace with −40 mV pre-pulse is shown in red. Graph shows voltage
dependence of slow inactivation of D21 and T21 cells. Inset, Half-maximal slow
inactivation potential. $n = 14$–52 cells. ***$P < 0.001$, two-tailed unpaired t-test or
Mann-Whitney test. The plot in A shows median, 25–75% interquartile range and
2–95% whiskers; bar charts and error bars in B-D depict means +/- SEM.

developmental stages and adulthood. Among these, in the DFC, pre-
sumably involving epigenetic mechanisms, we demonstrate that in
trisomy 21 we have an increase in gene dosage on hCh21 leading to
dysfunction in voltage-gated ion channels and disrupted excitatory
neurotransmission involving proteins expressed elsewhere in the
human genome, e.g. from chromosomes 1 (*KCND3*; *KCNC4*), 7 (*KCND2*)
and 19 (*KCNC3*). In future, further analyses of human transcriptomic
and proteomic datasets are likely to provide additional insights into
many of these pathways in Down syndrome.

### A-type K⁺ channels as a new therapeutic target for Down syndrome
Despite its high prevalence, we do not have credible therapeutic
strategies targeting the core neurological symptoms of Down syn-
drome. Most of the current unifying mechanistic hypotheses of neu-
rophenotypes revolve around broad developmental defects[3,13,14,16,17,48]

and GABAergic dysfunction[1,10,11,20,49]. Disruption of ion channel function
has been implicated in Down syndrome mouse models[7,50,51] and by
studying functional properties of human cells, for the first time, we
have identified dysregulation of many ion channel genes, one of which,
*KCND3*, underpins reduced A-type K⁺ currents in early trisomic cells.
Given the widespread expression of Kᵥ4.3 in neural somata and
dendrites[52] any alterations in ion channel function from an early
developmental stage will likely have important consequences, as also
indicated by our mathematical modelling of neural circuitry incor-
porating dysfunctional A-type currents. *KCND3* variants do cause
developmental phenotypes such as cognitive impairment, ataxia and
cardiac defects and there could be interesting parallels here with Down
syndrome[53,54]. Importantly, there are several new discoveries made in
this study regarding aberrant neural network activity, glutamate
synapses, Na⁺- and K⁺-channel dysfunction. Expression of TEA-
insensitive (eg – *KCND2*) and TEA-sensitive (eg – *KCNC3-4*) A-type K⁺
channels are also reduced post-birth in trisomy 21 suggesting that fast
transient A-type K⁺ channels may become a new therapeutically
tractable target for Down syndrome. Thus, translational approaches
designed to enhance the activity of these channels, and rescue the
aberrant phenotype using genetic or pharmacological strategies, while
ensuring that off-target and neurotoxic effects of K⁺ channel over-
expression are limited[55–57], will likely have potential benefits for Down
syndrome.

Taken together with the previously reported GABA-mediated
over-inhibition mediated mostly via GABA_A receptors[9–12] in Down
syndrome, down-regulation of glutamatergic synaptic activity, and
changes to voltage-gated Na⁺ and K⁺ ion channels indicate that epi-
genetic mechanisms are critically important for the Down syndrome
neurophenotype. It also further suggests that the excitation-inhibition
ratio may vary amongst Down syndrome people and across various
brain regions impacting on future therapies that will most likely need
to be more focused.

## Methods
### Induced pluripotent stem cells (iPSCs)
Two disomic (NIZEDSM1iD21-C3 and NIZEDSM1iD21-C9) and
trisomic (NIZEDSM1iT21-C5 and NIZEDSM1iT21-C13) induced
pluripotent stem cell (iPSC) lines described previously[18]. iPSCs were
generated at The Blizard Institute (QMUL), as described[18] primary
skin fibroblasts and used and stored under UK-Human Tissue Authority
License 12199, and ethical approval from the Ethical Committee of the
NELHA (North East London Health Authority), P/03/086. iPSCs were
cultured on Geltrex (ThermoFisher; A14133-02) coated wells in
Essential 8 (E8) medium (ThermoFisher; A1517001) supplemented
with penicillin/ streptomycin (ThermoFisher; 15140-122). Cells were
passaged in ReLeSR (Stem Cell Technologies; 100-0484) and 10 μM
ROCK inhibitor (Millipore-Sigma; Y0503) was included in culture
media for 24 hr post passage. iPSCs were validated for pluripotency as
previously reported[18,22].

### Neural stem cells
Neural stem cells (NSCs) were derived from isogenic iPSCs following
the Gibco protocol (Life Technologies, MAN0008031) and expanded
in neural expansion medium (NEM), consisting of Neurobasal (Ther-
moFisher; 21103049) and advanced DMEM/F12 (ThermoFisher;
12634010). The media was supplemented with neural induction sup-
plement (ThermoFisher; A1647801) and penicillin/ streptomycin. NSCs
were validated by immunolabelling and used for neuronal differ-
entiation between passages 6-10.

### Mouse astrocytes
Mouse astrocytes (ScienCell; SC-M1800) were expanded in AM astro-
cyte medium (ScienCell; SC-1831) supplemented with foetal bovine
serum (FBS; ThermoFisher; 0010), astrocyte growth supplement
(ScienCell; 1882) and penicillin/ streptomycin (ScienCell; 0503).

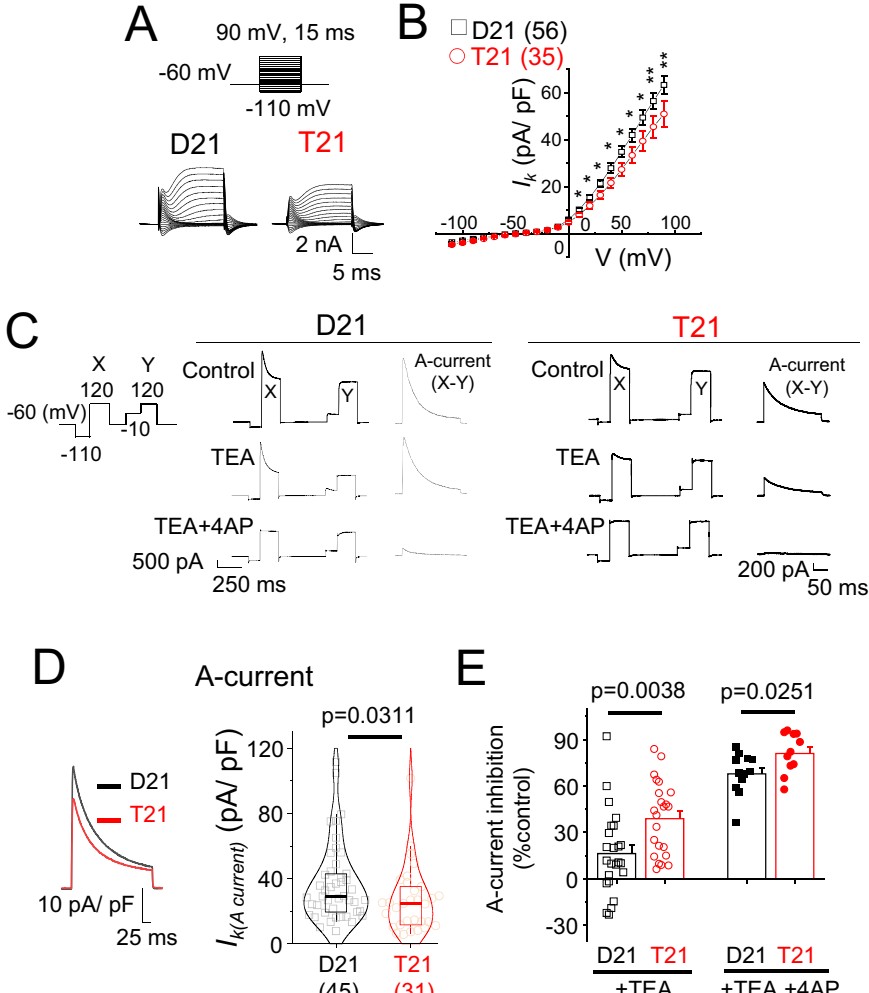

**Fig. 6 | A-type K⁺ channel deficits detected in early trisomy 21 neurons.**
**A** Voltage step protocol for constructing K⁺ channel I-V relationships along with example K⁺ currents of disomic (D21) and trisomic (T21) cells. **B** K⁺ current density I-V curves of D21 and T21 cells. **C** Step protocol for measuring A-currents. *Right*, A-current was measured by subtracting depolarisation at −10 mV (Y) from −110 mV (X). The same protocol was used in the presence of tetraethylammonium (TEA) followed by TEA and 4-aminopyridine (4AP) to measure A-current sensitivity to

these K⁺ channel antagonists. **D** Averaged A-current density waveforms and boxplot of A-current densities for D21 and T21 neurons. **E** Sensitivity of A-current density to TEA and 4-AP. Disomic cells are less sensitive to TEA than trisomic cells, a feature largely offset by co-applying 4-AP and TEA. $n = 11$–$56$ cells; two-tailed unpaired t-test or Mann-Whitney test. Box plots in D, show median, 25–75% interquartile range and 2–95% whisker; bar chart and error bars in (**B**, **E**) depict means +/- SEM.

## Neurons

For neuronal differentiation, isogenic NSCs were seeded onto glass coverslips and microelectrode array (MEA) plates (Axion biosystems; M384-tMEA-6W). Sterile coverslips in 24-well plates were coated with 400 μl of poly-L-ornithine (PLO: Millipore Sigma; P4957-50ML) at 37 °C overnight. PLO was removed and coverslips rinsed three times with Dulbecco's phosphate-buffered saline (DPBS; ThermoFisher; 14190-144) and coated with 20 μg/ml laminin (Millipore Sigma; L2020-1MG) for 3 hr at 37 °C. After laminin removal, coated coverslips were seeded with 30 ×10³ mouse astrocytes in AM media. 24 hr later, AM media was aspirated and 20 ×10³ isogenic NSCs were seeded on top of the mouse astrocytes in NEM. The next day, maintenance media made up of BrainPhys (StemCell Tech; 05793) supplemented with x1 SM-1 (Stem-Cell Tech; 05793), x1 N2A (StemCell Tech; 05793), 0.5% FBS (Biosera; FBS-016BS444-957247), penicillin-streptomycin (ThermoFisher; 15140-122), 1 mM cAMP (Millipore Sigma; D0627), 20 ng/mL BDNF (ThermoFisher; 45002), 20 ng/ mL GDNF (ThermoFisher; 45010) and ascorbic acid (Millipore Sigma; 49752-10 G) was applied to the cells. Maintenance media was changed twice per week and neurons differentiated for up to 180 days in vitro. MEA plates were coated in the same

way as the coverslips, but with a higher concentration of laminin (10 μg/ ml), and seeded with 60 ×10³ mouse astrocytes followed by 40 ×10³ isogenic NSCs.

Neuronal differentiation at sixty days in vitro was validated by immunolabelling. Neuronal differentiation was confirmed electrophysiologically by the presence of stereotypical action potentials that typify and confer neuronal identities. To cover for variability[43], we studied eight differentiations out of which inhibitory currents were detected in one differentiation and hence this batch was discarded from analysis. All our electrophysiological tests were sampled from at least three differentiations.

## Mycoplasma testing

All cells used were confirmed to be mycoplasma free by testing with the EZ-PCR Mycoplasma Detection Kit (Biological Industries, 20-700-20).

## Fluorescence in-situ hybridisation (FISH)

FISH on isogenic NSCs was performed as previously reported[22,58] with a XA 13/18/21 Probe (MetaSystems Probes; D-5607-100-TC). Over 500

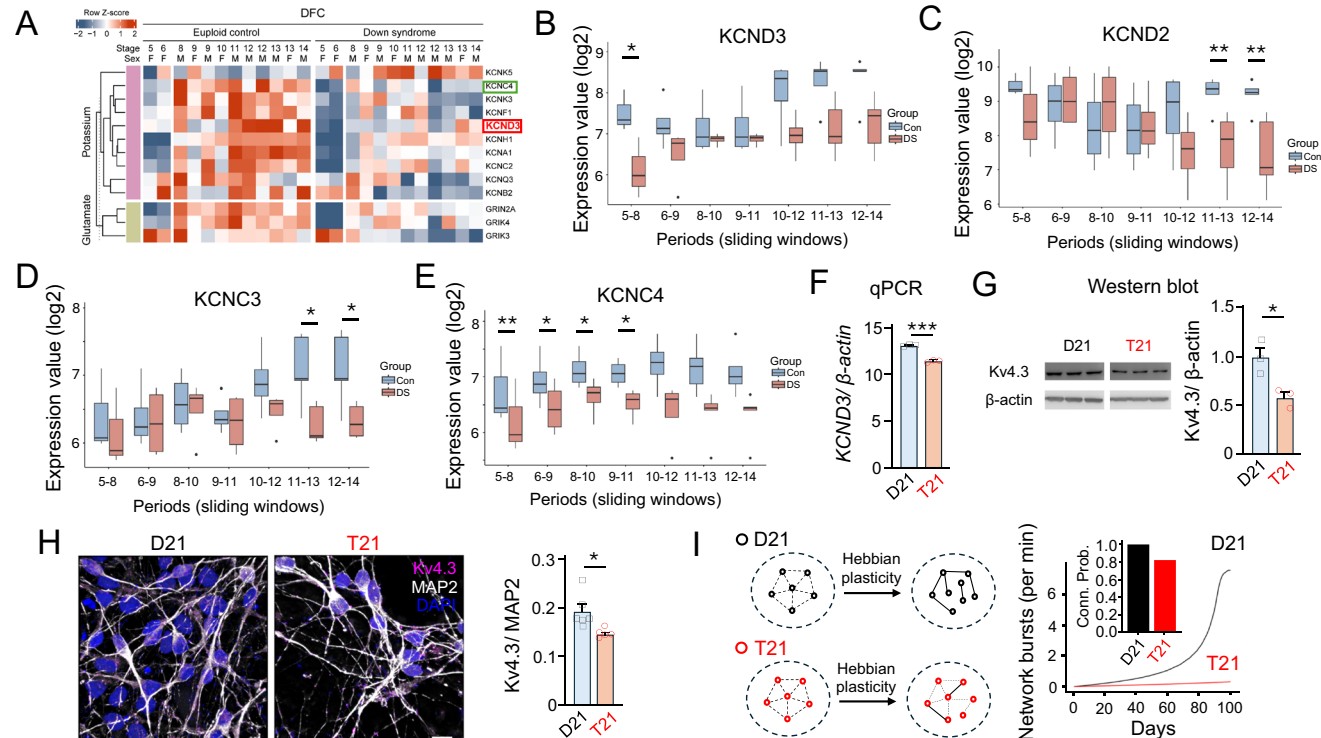

**Fig. 7 | Reduced expression of A-type K⁺ current genes and network bursting.**
**A** Metadata analysis of postmortem human RNA expression reveals expression changes (|Fold Change| > 1.3 and $P < 0.01$) in potassium channel-related (purple) and glutamate-related (green) genes in dorsolateral prefrontal cortex of Down syndrome tissues compared to euploid controls. Note that the expression of a TEA-insensitive A-type K⁺ channel Kv4.3 gene (red box) is reduced in Down syndrome but the other TEA-insensitive A-type channel genes, or HCN channels do not change at early developmental stages. Green box, reduced expression of a TEA-sensitive A-current channel gene *KCNC4*. **B−E** Expression of *KCND3* (**B**), *KCND2* (**C**), *KCNC3* (**D**) and *KCNC4* (**E**) across seven sliding windows corresponding to periods from mid-fetal development to adulthood (see Table S1). Boxplots depict median, 25−75% interquartile range and minimum and maximum values of gene expression. Note that A-current K⁺ channel gene expression is affected through life but the only TEA-insensitive A-current channel that is affected at stages 5-8 which corresponds to periods from mid-fetal development (16 weeks post-conception) to infancy (up to 6 months after birth) is *KCND3*. $n = 3−5$ brain samples. Two-tailed paired t-test.
**F** Normalised expression of *KCND3* mRNA in D21 and T21 isogenic iPSC-derived

neurons at 60 days in vitro. $n = 3$ samples. Two additional sets of primers gave similar results (Data S1). **G** Western blot gel images (left) and normalised expression levels (right) of Kv4.3 in D21 and T21 isogenic iPSC-derived neurons. $n = 3$ samples (uncropped gels in Figure S9A). (**H**), Representative images (left) and normalised expression levels (right) of Kv4.3 in the same neurons using immunocytochemistry. $n = 5$ (D21) and $n = 6$ (T21) independent 3D image stacks (**I**), Computational model relating cellular effects of Kv4.3 currents to network bursts. *Left*, Schematic showing two networks undergoing Hebbian plasticity for multiple days. The networks differ by the intrinsic features that control the propensity of generating intrinsic bursts. Hebbian learning enhances small differences in bursting propensity with time, giving more connections and more net bursting as plasticity unfolds. T21 (red) is modeled by a reduced propensity for bursting with respect to D21 (black). *Right*, Network burst as a function of days. *Inset*, Network connectivity in D21 (1.0) and T21 (0.81). *$P < 0.05$, **$P < 0.01$; *$P < 0.05$, **$P < 0.01$, ***$P < 0.001$. Scale bar = 10 μm. For **F−H** data are presented as mean values +/- SEM with analyses performed using either paired or unpaired two-tailed t-tests.

nuclei from eight different Z-stacks were evaluated for each clone. Based on the observed number of fluorescent hybridisation signals, nuclei were assigned to four different categories, namely "one signal", "two signals" "three signals" and ">three signals". Damaged nuclei or overlapping nuclei were excluded from scoring.

## Microelectrode array recordings
Simultaneous microelectrode array recordings were carried out partially blinded to the genotype (two out of five differentiations; blinded differentiations were used after unblinded recording to confirm the results) in maintenance media from 64 low-noise electrodes (electrode diameter 50 μm; arranged as an 8 ×8 array with 300 μm spacing) per well using a Maestro Pro (Axion Biosystems) system for 10 min at 37°C. For pharmacological experiments, 0.5 μM tetrodotoxin or 10 μM CNQX and 25 μM APV was added directly to the recording media. Data was sampled at 12.5 kHz, digitized and analysed using AxIS Navigator (ver 1.5.2; Axion Biosystem) and Neural Metric Tool (ver 2.4.12; Axion Biosystem). Spikes were detected using an adaptive detection threshold of 6 standard deviations over baseline noise for each electrode with 1 s binning. Active electrodes were defined as those that detected a

minimum of five spikes per minute. For weighted mean firing rates, only electrodes whose activity was greater than active electrodes were considered. Single electrode bursts were detected with a maximum of 100 ms inter-spike interval and a minimum of five spikes per burst. Network bursts were detected with the same maximum inter-spike interval but a minimum of 50 spikes and 20-35% electrodes participating in bursting synchronously within 20 ms windows. Area under normalised cross-correlation of network burst was calculated from the area under pooled inter-electrode cross-correlation. Synchrony index was calculated as previously described[59]. Synchrony index takes a unitless measure of synchrony between 0 and 1 with values closer to 1 indicating higher synchrony.

## Whole cell electrophysiology
**Extracellular solution.** Whole cell electrophysiology of iPSC-derived neurons was carried out partially blinded of the genotype (two out of seven differentiations included; blinded differentiations were used after unblinded recordings to confirm the results) using borosilicate glass patch electrodes (resistances of 3-5 MΩ) in a saline solution containing (mM): 140 NaCl, 4.7 KCl, 1.2 MgCl₂, 2.52 CaCl₂, 11 glucose,

and 5 HEPES, pH 7.4. For experiments in zero $Mg^{2+}$, $MgCl_2$ was removed from the saline. All drugs were applied through a modified U-tube[60]. AMPA, NMDA and $GABA_A$ receptor current densities were calculated by dividing the agonist-activated current by whole cell capacitance of individual cells.

**Internal solutions.** Excitatory postsynaptic currents (EPSCs) were recorded using an internal solution previously described[28] containing (in mM): 145 Cs methanesulfonate, 5 MgATP, 10 BAPTA, 0.2 $Na_2$GTP, 10 HEPES, 2 QX314 and pH − 7.2 or a CsCl internal[29] containing (mM): 120 CsCl, 1 $MgCl_2$, 11 EGTA, 30 KOH, 10 HEPES, 1 $CaCl_2$, and 2 $K_2$ATP, pH 7.2. $Na^+$ currents, whole cell GABA, AMPA and NMDA currents as well as inhibitory postsynaptic currents (IPSCs) were also measured using the CsCl internal. Action potentials were recorded using a K-gluconate[28] internal solution containing (in mM): 130 K-gluconate, 10 NaCl, 2 $Na_2$ATP, 0.3 NaGTP, 10 HEPES, and 0.6 EGTA, buffered to pH 7.4 or a K-methanesulfonate internal containing (in mM): 140 $KMeSO_4$, 5 EGTA, 0.1 $CaCl_2$, 2 Mg-ATP, 4 $Na_3$-GTP, 10 HEPES, and pH at 7.3. $K^+$ currents including A currents were recorded using the K-gluconate internal[61] and HCN voltage sags were recorded in the K-methanesulfonate internal[62]. The osmolarity of all internal solutions were in the range 275-300 mOsm.

**EPSCs.** EPSCs were recorded in voltage clamp, at a holding potential of −70 mV. Here, cells were considered as neurons if they received EPSCs in $Mg^{2+}$ saline or $Mg^{2+}$-free saline or if they had bursting activity in $Mg^{2+}$-free saline. Around 99% of cells were neuronal in these recordings according to these criteria. After establishing whole cell configuration, a −10 mV pulse was used to construct a capacitance discharge curve and whole-cell capacitance was calculated from the area under the capacity discharge[29]. EPSCs were identified from their characteristic faster rise and decay kinetics compared to IPSCs[29,63] and/ or from their sensitivity to blockage by the AMPA receptor antagonist CNQX (10 μM) and NMDA receptor antagonist APV (25 μM). EPSC frequency and amplitudes were measured from 2-5 min recording epochs. EPSCs kinetics were measured from uncontaminated events from cells that yielded enough clean events. Rate of rise and decay times were reported for average waveforms that were reliably fit using a line equation for rate of rise and combined mono- and bi-exponentially fitted weighted decay times according to the following equation:

$$t_w = (A1.t1 + A2.t2)/(A1 + A2) \qquad (1)$$

where $t_w$ is the weighted decay time and t1, t2 and A1, A2 are the decay times and areas of the first and second decay time components.

Bursts in voltage clamp were classified by clear deflections of membrane currents over baseline ( > 8 pA) lasting for at least 2 s. Burst amplitudes were measured as peak membrane currents devoid of AMPA EPSCs and durations were times taken by the membrane currents to return to baseline.

Charge transfer during bursts was calculated as the area under the bursting current waveform including AMPA EPSCs that were present during the burst.

Voltage clamp recordings were carried out with optimised series resistance ($R_s$, < 10 mΩ) and whole-cell membrane capacitance compensation. Membrane currents were filtered at 5 or 10 kHz ( − 3 dB, sixth pole Bessel, 36 dB per octave).

**Action potentials.** Action potentials were recorded in the $Mg^{2+}$ containing saline solution. The resting membrane potential of each cell was noted immediately ( < 10 s) after establishing robust whole-cell configuration in the absence of any current injection. Initial resting membrane potentials were not recorded for cells that took >10 s to achieve adequate whole-cell stability. This was followed by construction of a capacitance discharge by pulsing −10 mV voltage steps from

which whole-cell capacitance, membrane time constant and input resistance of the cells was calculated. Next, the basal spiking activity of neurons was monitored in current clamp for at least 0.5 −1 min. Most cells were studied at resting membrane potential, and no current was injected. Where required, neurons were held close to −60 mV by injecting small hyperpolarizing currents. Subsequently, cells were depolarized by injection protocol of a step protocol with 5, 10 or 25 pA increments of currents. Cells were categorized as neurons if they fired action potentials during monitoring of spontaneous spiking activity or during administration of the step protocol. The step current protocol was used to calculate the current at which maximum spikes were elicited and to estimate the rheobase and identify spiking properties of neurons. Neurons were categorized as single, double or multiple spikers if spiking saturated at one, two or greater than two spikes. The rheobase estimate derived from the current step protocol was refined and measured at pA precision by injecting single current injections. After establishing accurate rheobase, an input-output curve of spike firing was generated for x1-4 increments of depolarising rheobase step injections. Input-output curves were averaged from at least two-five runs of the rheobase protocol. Average latency was measured as the time at which the first spike appeared after the beginning of the x1 rheobase current injection and spike jitter was calculated as the co-efficient of variation of latency from cells that withstood at least five runs of the rheobase protocol. When the rheobase changed before the administration of the protocol five times, the rheobase and input-output curves at the lower current was reported. In these cases, the rheobase was measured again and latency and jitter were only reported if the rheobase did not change too dramatically ( > 5 pA) and if the protocol was executed at least five times for calculation of average latency and co-efficient of variation. Spike kinetic parameters and threshold for action potentials were measured from the first spike at rheobase.

**$Na^+$ currents.** $Na^+$ currents were recorded in $Mg^{2+}$ containing saline supplemented 1 mM $BaCl_2$, 1 mM 4-AP, 1 mM TEA and 200 μM $CdCl_2$ to block $K^+$ and $Ca^{2+}$ channels. $Na^+$ currents densities were constructed by subtracting averaged currents (ten sweeps) generated by stepping from −60 to 20 mV in control and in the presence of 0.5 μM tetrodotoxin, measuring the peak current and dividing by whole cell capacitance of the cells. Similarly, $Na^+$ I-V curves were generated at a holding potential of −60 mV by stepping from −80 to 55 mV in 5 mV increments in control and tetrodotoxin, subtracting the waveforms, measuring peak currents at each voltage step and calculating current densities by dividing whole cell capacitance of the cells. Inactivation was measured by subtracting currents in control and tetrodotoxin while stepping from −80 to 10 mV in 10 mV increments from −120 mV (100 ms) following a brief −10 mV test pulse. For fast and slow inactivation, $Na^+$ channels were activated for 100 ms and 5 s respectively during the prepulse stage followed by stepping to 0 mV for 10 ms and hyperpolarization to relieve inactivation at −120 mV for 100 ms before returning to holding potentials of −60 mV. Normalised current curves immediately after pre-pulsing were generated and fitted with an inhibition curve according to the Boltzmann equation:.

$$I(v) = I_{max}/(1 + \exp((V_{50} - V)/km)) \qquad (2)$$

where $I_{max}$ is maximum current, $V_{50}$ is the voltage where I(v) is 50% of $I_{max}$ and km is the slope.

**$K^+$ currents.** $K^+$ channel currents were recorded in the $Mg^{2+}$ containing saline supplemented with 0.5 μM tetrodotoxin and 200 μM $CdCl_2$ to block $Na^+$ and $Ca^{2+}$ channels. $K^+$ channel I-V curves were generated by measuring steady state currents while stepping from −110 to 90 mV in 10 mV increments from a holding potential of −60 mV and factoring in whole cell capacitance resulting from the area under a capacity

discharge curve. Transient A-type $K^+$ currents were measured by averaging ten sweeps of a voltage protocol while alternating a 125 ms pre-pulse of −110 or −10 mV to 120 mV for 200 ms. The transient current density was digitally obtained by subtraction of the −10 to 120 mV transition from the −110 to 120 mV step and dividing by the capacitance. These alternating steps were repeated in the presence of 1 mM tetraethylammonium (TEA) followed by 1 mM TEA and 3 mM 4-aminopyridine (4AP) to obtain TEA-sensitive and TEA-and-4AP-sensitive A-current densities.

A-current activation profiles were measured using a voltage protocol that delivered test potentials positive to −60 mV and up to +150 mV (10 mV increments; 100 ms) immediately following a pre-pulse at −110 mV for 500 ms. A-current inactivation profiles were recorded using a voltage protocol by pre-pulsing from −120 to 10 mV (10 mV increments; 100 ms) followed by a test pulse at +40 mV for 200 ms.

**HCN channels.** HCN voltage sags[64,65] resulting from hyperpolarising current injection steps were measured from disomic and trisomic cells. In order to study the maximum size of sags that resulted from the lowest membrane potential (or the highest possible hyperpolarising current achievable), currents were carefully injected in suitable equal increments such that the maximum hyperpolarising step yielded membrane potentials close to −150 mV or the cell underwent di-electric breakdown in which case the membrane voltage measures in the sweeps immediately prior to di-electric breakdown were analysed. The sizes of the sags for the increments of currents have been plotted and compared between disomic and trisomic cells.

### Immunolabeling and fluorescence imaging

Immunostaining was performed as previously reported[22,23] using primary antibodies against MAP2 (Abcam; ab5392; dilution – 1:1000), SMI-312 pan-axonal neurofilament marker cocktail (Biolegend; 837904; 1:1000), PSD95 (Cell Signaling; 3450; 1:400), Synapsin-1 (Cell Signaling; 5297; 1:400), TUBB3 (Biolegend; 802001; 1:5000), Kv4.3 (Proteintech; 25468-1-AP; 1:200), Kv4.2 (Proteintech; 21298-1-AP; 1:200), VGLUT1 (Synaptic Systems; 135 302; 1:1000), Nestin (Abcam; ab6320; 1:200), PAX6 (Biolegend; 901301; 1:200) and SOX2 (Santa Cruz; sc-17320; 1:200). Primary antibodies were labeled with goat anti-chicken Alexa Fluor 633 (ThermoFisher; A-21103; 1:500), donkey anti-mouse Alexa Fluor 488 (ThermoFisher; A-21202; 1:1000), donkey anti-rabbit Alexa Fluor 555 (ThermoFisher; A-31572; 1:1000), donkey anti-goat Alexa Fluor 555 (ThermoFisher; A-21432 (1:1000), donkey anti-rabbit Alexa Fluor 488 (ThermoFisher; A-21206; 1:1000), donkey anti-mouse Alexa Fluor 647 (ThermoFisher; A-31571; 1:500), donkey anti-mouse Alexa Fluor 555 (ThermoFisher; A31570; 1:1000), donkey anti-rabbit Alexa Fluor 647 (ThermoFisher; A-31573; 1:500) secondary antibodies.

Regions of interest were chosen close to rosettes and images were acquired in 3D stacks using an Olympus FV 3000 inverted confocal microscope with x60 oil-immersion objective. Image analysis was performed using Imaris (ver 9.9.1; Oxford Instruments) on two disomic and two trisomic isogenic clones using "surface" option. The total surface of each synaptic marker was normalized to the total cell surface staining for MAP2 in Z-stacks. Expression was analysed in all MAP2-positive neurons, in the perikaryon and dendrites. The proportion (%) of VGLUT1 positive neurons was measured by masking MAP2 positive neurons and quantifying the number of VGLUT positive and negative cells.

### Quantitative PCR (qPCR)

Total RNA was isolated at 60 days in vitro using the RNeasy Mini kit (Qiagen; 74104) according to manufacturer's instructions and cDNA was prepared using high-capacity RNA-to-cDNA Kit (Applied Biosystems; 4368814). qPCR was performed using the PowerTrack™ SYBR Green Master Mix for qPCR (Applied Biosystems; A46110) on a 7500 Real-Time PCR system (Applied Biosystems) using one set of primers for β-actin (F – GGACTTCGAGCAAGAGATGG, R – AGCACTGTGTTGGCGTACAG) and three sets of primers for KCND3 (Set 1: F – TCCACCATCAAGAACCACGA, R – TCTTACTACGACGG-GAGCAG; Set 2: F – CACAAGCATCCCTGCCTCGTTT, R – TCAAGGAG-CAGATGGAGCCGAA); Set 3: F – CATGACCACACTGGGATACG, R – CAATCACAGGGACTGGCAG). Gene expression was calculated using the ΔCT method.

### Western blot

Whole-cell lysates were prepared in RIPA Lysis Buffer (Millipore; 20-188) buffer with phosphatase (ROCHE, 04 906 837 001) and protease inhibitors (ROCHE; 11836170001). Bradford Assay Reagent (Thermo-Fisher; 1863028) was used for quantification of protein concentrations. The proteins were separated in a NuPAGE™ 4–12% Bis-Tris Gel (Invitrogen; NP0323BOX) and transferred to a nitrocellulose membrane according to the manufacturers protocols (Bio-Rad). The membranes were incubated with the same primary antibodies as immunolabelling studies for Kv4.3 and Kv4.2 but at a concentration of 1:1000 along with β-actin (Sigma-Aldrich; A5441; 1:60000) followed by HRP-conjugated secondary antibodies (goat anti rabbit, Abcam, ab6721, 1:20000; goat anti mouse, Abcam, ab6728, 1:2000). The bands were visualized with SuperSignal™ West Femto Maximum Sensitivity Substrate (Thermo Scientific; 34095) while quantification was carried out using BioRad software and density was calculated in ImageJ (Fiji ver using 1.54p) using the "gel analysis" option. All signals were normalised to corresponding β-actin.

### RNA expression metadata analysis

**Data preprocessing.** The series matrix file of GSE59630 from NCBI GEO was downloaded and used for analysis. Exon IDs were converted to official gene symbols using the gene ID conversion of Database for Annotation, Visualization and Integrated Discovery 2021 (DAVID 2021) to label the genes in figures and tables[66].

**Differentially expressed gene analysis.** Differentially expressed gene (DEG) analysis between Down syndrome and matched control samples was carried out partially blinded (without prior knowledge of A-current defects that were detected using electrophysiology) during all development periods, using a paired t-test. Human brain developmental periods were based on criteria previously described[67]. In addition, a sliding window approach for developmental stages was used to diminish the uniqueness of each sample and emphasize the temporal pattern. Three developmental stages were grouped and are displayed together. A paired t-test was used to estimate the significance of expression differences between Down syndrome and control samples at each sliding window. Genes with a fold-change ≥ 1.3 and p < 0.01 were considered significantly changed. Where necessary, *p* values were corrected using the false discovery rate (FDR) for multiple comparisons.

### Mathematical modelling

We created a network of 100 principal cells and modeled rate-based neurons having a propensity for generating spontaneous bursting according to a Poisson process with rate λ. We consider that this propensity depends on the net input to each unit such that

$$\lambda_i = \lambda_0 + \beta^* \sum_j w_{ij}(d)\, r_j \qquad (3)$$

where each neuron $i$ receives a weighted sum of the firing rate $r_j$ of all other neurons, labeled by index $j$. The burst propensity in the absence of inputs, $\lambda_0$, is taken to be 0.1 for modeling D21 and 0.06 for modeling T21. This propensity to burst is thought to be at least partially controlled by Kv4.3-like currents[41,42]. The parameter β controls how the net

input to a neuron relates to burst production and it was set to 0.01. Poisson processes were simulated based on a time-step of 1 min such that rates in units of bursts per min. For each simulated run over a day, the simulated time was set to 24 hr.

The connectivity is modeled as a matrix $w_{ij}$ which changes with the number of simulated days $d$. This connectivity is initiated as uniformly zero and then follows a Hebbian learning rule $\Delta w_{ij} = \alpha r_i r_j$ with weights bounded between 0 and 1. The Hebbian learning rule is applied once per simulated day and the learning rate $\alpha$ is set to 0.000345. The connection probability is calculated as the fraction of weights that are larger than 0.2. A network burst is calculated as the rate at which the net input to a neuron exceeds 0.1. The learning step and network simulation step were simulated in alternation for 100 simulated days.

All associated codes can be assessed from: https://github.com/nauralcodinglab.

## Statistical analysis

Recordings were analysed using Clampfit (10.6; Molecular Devices), WinEDR (ver 4.0.2; Strathclyde Electrophysiology Software) and WinWCP (ver 5.7.0; Strathclyde Electrophysiology Software). Given the inherent variability of iPSC-derived neurons[43], large sample sizes were selected based on previous experience. All data, when applicable, were checked for normal distributions using Kolmogorov-Smirnov test in Instat (ver 3.06; Graphpad) or Prism (ver 9; Graphpad). Normally distributed data were compared using a two-tailed unpaired t-test or one-way ANOVA and non-parametric results were compared using a Mann-Whitney test using Instat or Prism. Plots were generated and curves fitted using Origin 6 or OriginPro 2021b (Originlab). Violin plot bar graphs represent mean ± standard error of mean and boxplots depict median, 5-95% and interquartile ranges. Where applicable, individual data points have been depicted on plots and numbers in brackets show n-number of cells or samples. Data S1 contains all data contained in the manuscript along with the statistical tests and *P* values.

**Materials availability**. Materials used in the manuscript will be made available upon request by the corresponding authors under suitable materials transfer agreements (MTAs).

## Reporting summary

Further information on research design is available in the Nature Portfolio Reporting Summary linked to this article.

## Data availability

The source data are provided with the Source Data file. Source data are provided with this paper.

## Code availability

All associated code can be accessed from: https://github.com/neuralcodinglab.

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

## Acknowledgements

This work was funded by a Wellcome Trust Collaborative Award in Science 217199/Z/19/Z. AM was funded by a William Harvey Academy Fellowship, co-funded by the People Programme (Marie Curie Actions) under REA no. 608765. SBH was awarded a fellowship from International Rett Syndrome Foundation (3606). AP and IA were funded by Adris foundation, NPOO-NextGeneration (NEURO-MORF) and the Croatian Science Foundation (HRZZ-UIP-2025-02-5828), CRP-ICGEB (CRP/HRV25-03).

## Author contributions

Conceptualisation–S.B.H., D.N., T.G.S. Electrophysiology–S.B.H., M.M. Confocal Imaging–I.A., A.P. Differentiation of iPSC cultures–A.M., I.A., A.P., P.A.G., N.L.O'B. Maintenance of cultures-A.M., I.A., A.P., P.A.G., N.L.O.B., S.B.H. Transcriptomics analysis–J.K., H.J.K. Mathematical modelling–R.N. Funding acquisition and project leadership–S.B.H., D.N. and T.G.S. The initial draft was written by S.B.H. and T.G.S., and all authors contributed to the writing of the manuscript.

## Competing interests

The authors declare no competing interests.
