## [Transparent Peer Review file · Nature Communications]

Synaptic and intrinsic membrane defects disrupt early neural network dynamics in Down syndrome

Corresponding Author: Professor Trevor Smart

Version 0:

Reviewer comments:

Reviewer #2

(Remarks to the Author)

The manuscript by Hannan et al entitled "Synaptic and intrinsic membrane defects disrupt early neural network dynamics in Down syndrome" explored the developing neural network dynamics of a cellular model of trisomy 21 based on neurons derived from induced pluripotent stem cell. They used a very nice and appropriate cell model deriving from the same individual but exhibiting mosaicism regarding the trisomy genotype. This cellular model is perfectly suited to this study. In the past, mouse models have been used, but they failed to reproduce all the genetic alterations observed in the disease. In this study, the authors use a human cellular model, which provides a highly suitable framework. What's more, they compile expressions at both neuronal network and neuronal scale, and provide patient data to corroborate their findings. In particular, they identify a KCND3 gene encoding the Kv4.3 potassium channel which is altered and thus provides a potential new therapeutic target for trisomy 21.

Authors investigated network activity using multiple electrode array (MEA) as well as individual synaptic and intrinsic properties. They identified several neurophysiological alterations such as deficit in excitatory synaptic transmission and in ionic conductance associated with spike waveforms. Overall they accomplished a lot of work and very thorough investigation. These results will have a great impact on the field and on the society. However, to strengthen the manuscript, I have only few comment that I would like the authors to address.

1. Table S1 and DataS1 were not provide in the material for review.
2. The authors started the report with two clonal lines for euploid control and aneuploid trisomy 21 and show no difference in network dynamics using MEA. It is not clear in the rest of the manuscript if they work with all lines of selected one of each type.
3. Authors declared that they performed eight differentiations in excitatory neurons but one failed and gave some inhibitory neurons. They have discarded this latter differentiation. They have done a lot of recording from only 7 differentiations and it is not clear if each given data set derived from one of several differentiations. Authors should clarify.
4. Quantification of synaptic markers PSD95 and synapsin versus MAP2. How the quantitative analysis of fluorescence image has been done? It is said in method that it has been normalized to MAP2 signal but how. What are the region of interest? In panel K, it looks 11 neurons per conditions but the legends indicate 11-59 cells/3D image stack. Please clarify.
5. One of the conclusion of the authors is that These results
6. Authors suggest that trisomic neurons exhibit a deficiency in glutamatergic synaptic connectivity without notable changes to cell surface glutamate receptor. However, they based these data on spontaneous activity and local application of agonist. The classical approach would be to analyze the properties of miniature ESPC. Furthermore, they asses the amount of cell surface receptor to the amplitude of the current evoked by a local application of AMPA of NMDA on very few cells. This is not satisfactory.
7. Line 801-802 in figure 4, there is a mismatch between the lettering of the panels on the figure and the lettering in the legend. In panel F, how many waveforms have been averaged?
8. Line 218-219. The sentence regarding the conclusion of TEA sensitivity is not clear. Authors show that there is less K⁺ A current but also show higher inhibition by TEA. Their experiments suggest a switch in K⁺ current component.
9. Immunolabeling section. Authors should include the dilution of antibodies.
10. Line 520-521, please provide the concentration of cAMP, GDNF and BDNF.

Reviewer #3

(Remarks to the Author)

In this manuscript the authors characterize intrinsic and network properties of neuronal cultures derived from iPSCs with normal chromosomal complement and with trisomy 21. Compared to controls glutamatergic synaptic activity is reduced (particularly AMPA mediated currents). Neurons with trisomy 21 show reduced excitability and deficits in A-current signaling

Overall assessment

This is a carefully conducted study addressing an important and timely issue in understanding Down Syndrome pathology. With increasing longevity of Down Syndrome patients, a better understanding of the neurological and cognitive aspects becomes ever more urgent and identification of potential drug targets more pressing. As detailed in the questions and comments, the usefulness in this approach heavily depends on the faithfulness with which the in-vitro approach recapitulates early development in-vivo. It would be useful for authors to expand on this a bit more in the manuscript's discussion. The lack of GABAergic activity should ideally also be discussed a bit more in detail – given that this is one of the more widely reported areas of functional defect in Down Syndrome patients.

Main questions and comments :

1. Can the authors expand on the faithfulness with which iPSC derived neurons recapitulate embryonic development? Especially the absence of GABAergic neurons from most of their cultures seems intriguing, since other attempts in this direction observed inhibitory GABAergic activity.
2. Line 566 - Whole cell electrophysiology of iPSC-derived neurons was carried out partially blinded of the genotype (two out of seven differentiations included). Can the authors clarify this statement? How many differentiations were blinded? Were the unblinded investigations done first?
3. Given that human transcriptomic data is available, what was the reason for the authors to not extend the analysis to a genome-wide comparison and to a comparison between iPSC derived neurons and neurons obtained from human cortex?

Detailed questions and comments:

1. "...and also because early-stage iPSC-derived neurons and immature neurons display developmentally regulated slow Ca²⁺ oscillations. Thus, MEA recordings ensured that only action potential-dependent changes to excitability were analysed". While MEA recordings might have been a sensible choice here, why is their inability to detect slow calcium oscillations a methodological advantage?
2. Line 136. "As expected, in the absence of Mg²⁺, neurons showed greater synaptic activity but 10% of trisomic cells (compared to 0% disomic neurons) did not exhibit synaptic currents despite undergoing 0 Mg²⁺-induced membrane bursting (Figure 2F) thereby identifying a proportion of cells that fail to integrate into cortical networks during development." These findings would indicate an abundance of silent (NMDA only synapses). Would this also indicate a deficit in synaptic plasticity? Could such a deficit be causal for the reduced synaptic integration?
3. What is the reason the cumulative data for Spontaneous activity in Fig2 is shown as violin plots and not standard cumulative histograms? These are easier to quantitatively compare and analyze.

Reviewer #4

(Remarks to the Author)

In the manuscript "Synaptic and intrinsic membrane defects disrupt early neural network dynamics in Down syndrome", Hannan et al used isogenic pairs of Down syndrome (DS) iPSC-derived neurons to identify early neurodevelopmental deficits. Using multielectrode array (MEA) and patch-clamp electrophysiology, the authors show that trisomy 21 (T21) neurons have reduced glutamatergic synaptic connectivity, altered Na⁺ and K⁺ channel function, impaired spike firing, and disrupted network synchrony. The authors further conclude downregulation of Kv4.3 in T21 excitatory neurons could be the underlying mechanism.

This work is important to the field. However, the manuscript presented in the current form needs to be substantially improved to support the conclusions.

Major concerns:

1. The data characterizing neurons derived from both Trisomic 21 (T21) and isogenic Disomic 21 (D21) iPSCs should be included. The authors show neural stem cell (NSC) markers (e.g., SOX2, Nestin, PAX6), general neuronal markers (e.g., TUBB3, MAP2), and synapse marker staining. However, to show that their differentiated neuronal differentiation cultures give rise predominantly to glutamatergic neurons, the authors should also include immunostaining for glutamatergic markers (e.g., VGLUT1 and VGLUT2 for excitatory neurons) as well as inhibitory neuron markers (e.g., GABA or GAD65/67).
2. The data to support the mechanism on KCND3 (Kv4.3) downregulation in T21 neurons is insufficient. KCND3 mRNA expression data shown in Figure 7 is obtained from post-mortem brain datasets. What is the expression profile of KCND3 and associated genes in the iPSC-derived neurons reported in this work? Without direct confirmation in these iPSC-derived neurons, it is difficult to draw conclusion. Additional verification should also be provided to show that the decreased mRNA level of KCND3 leads to reduced protein expression and impaired K⁺ channel membrane trafficking. Hence, additional experiments the authors should consider include (i) RNA-seq or qPCR in their iPSC-derived neurons to show that KCND3 mRNA is indeed decreased. (ii) Western blot and/or immunofluorescence to show Kv4.3 protein expression in T21 iPSC-derived neurons.

3. Functional rescue data on Kv4.3 is lacking. For instance, would overexpression of KCND3 rescue the Kv4.3 phenotype in T21 neurons? Would a relatively selective Kv4.3 opener rescue the phenotype observed in T21 neurons?
4. In the discussion, the authors should acknowledge that their work is based on a simplified in vitro culture system composed almost exclusively of glutamatergic neurons (upon showing immunofluorescence staining of VGLUT1, etc, as suggested earlier). This is important as the simplified system is different from co-culture or in vivo animal models, where both inhibitory and excitatory neurons are present, among other cell types.

Minor points:

1. In Figure 3, Panel B, inset is only included for D21, but not for T21. Please add inset for T21.
2. In the Figure 3 legend, Line 801-802 seems to describe panels N, O, P, Q, which are not shown in Figure 3. Please double check.

Version 1:

Reviewer comments:

Reviewer #2

(Remarks to the Author)

The authors responded satisfactorily to all my questions and comments. This manuscript addresses issues that are very important for understanding the pathology of Down syndrome. I suggest this manuscript to be accepted for publication.

Reviewer #3

(Remarks to the Author)

The authors have addressed the points I have raised during the initial review. I have no further open questions.

Reviewer #4

(Remarks to the Author)

The revised manuscript has addressed all of my concerns, except for the issue regarding the functional rescue data on Kv4.3. The rationale provided for not performing a rescue experiment is unconvincing, and the discussion of this point in the main text remains insufficient. Therefore, I would still recommend including functional rescue data in this manuscript.

Reviewer #2 (Remarks to the Author)

The manuscript by Hannan et al entitled “Synaptic and intrinsic membrane defects disrupt early neural network dynamics in Down syndrome” explored the developing neural network dynamics of a cellular model of trisomy 21 based on neurons derived from induced pluripotent stem cell. They used a very nice and appropriate cell model deriving from the same individual but exhibiting mosaicism regarding the trisomy genotype. This cellular model is perfectly suited to this study. In the past, mouse models have been used, but they failed to reproduce all the genetic alterations observed in the disease. In this study, the authors use a human cellular model, which provides a highly suitable framework. What's more, they compile expressions at both neuronal network and neuronal scale, and provide patient data to corroborate their findings. In particular, they identify a KCND3 gene encoding the Kv4.3 potassium channel which is altered and thus provides a potential new therapeutic target for trisomy 21.

Authors investigated network activity using multiple electrode array (MEA) as well as individual synaptic and intrinsic properties. They identified several neurophysiological alterations such as deficit in excitatory synaptic transmission and in ionic conductance associated with spike waveforms. Overall they accomplished a lot of work and very thorough investigation. These results will have a great impact on the field and on the society. However, to strengthen the manuscript, I have only few comment that I would like the authors to address.

We would like to thank the reviewer for finding our work “thorough”, using a “very nice and appropriate cell model” that “provides a highly suitable framework” that will have “a great impact on the field and on the society”.

All concerns raised have now been addressed in the revised version of the manuscript (see blue text) which further strengthen the conclusions of our study.

1. Table S1 and DataS1 were not provide in the material for review.

Our apologies, Table S1 and Data S1 are now attached to the manuscript and available for review.

2. The authors started the report with two clonal lines for euploid control and aneuploid trisomy 21 and show no difference in network dynamics using MEA. It is not clear in the rest of the manuscript if they work with all lines of selected one of each type.

We apologise for the confusion. In the initial version of the manuscript, we did not find any difference between network dynamics of the two disomic (C3 and C9) or trisomic (C5 or C13) lines by screening multiple MEA parameters (including, no. of active electrodes, weighted mean firing rate, number of bursts, number of network bursts and network synchrony; Figure S2A). Therefore, for the rest of the manuscript, in all measurements we used both disomic and trisomic clones to report on the other parameters by pooling results together. This has now been discussed in the manuscript.

3. Authors declared that they performed eight differentiations in excitatory neurons but one failed and gave some inhibitory neurons. They have discarded this latter differentiation. They have done a lot of recording from only 7 differentiations and it is not clear if each given data set derived from one of several differentiations. Authors should clarify.

To clarify, all our differentiations gave consistent results except for one, as noted by the reviewer. This type of aberrant differentiation of iPSCs has been observed previously and we discarded this one differentiation to err on the side of caution as stated in the manuscript. All our physiology datasets contain results from at least three differentiations (eg- MEA was

acquired from five differentiations) and these details have now been added to the methods section of the revised manuscript.

4. Quantification of synaptic markers PSD95 and synapsin versus MAP2. How the quantitative analysis of fluorescence image has been done? It is said in method that it has been normalized to MAP2 signal but how. What are the region of interest? In panel K, it looks 11 neurons per conditions but the legends indicate 11-59 cells/3D image stack. Please clarify.

To quantify the levels of synaptic makers we normalized staining for PSD95 and Synapsin-1 to MAP2. ROIs were chosen close to cell rosettes and analyses performed with Imaris 9.9.1 software. The total surface of each synaptic marker was normalized to the total surface staining for MAP2 per figure (Z-stack). Expression was analysed in all MAP2-positive neurons, including the perikaryon and dendrites. In panel K, the 11 data points represent 11 Z-stack figures, each containing multiple neurons. These details have now been added to the revised version of the manuscript.

5. One of the conclusion of the authors is that These results

We are unsure what is being requested here, the text seems truncated

6. Authors suggest that trisomic neurons exhibit a deficiency in glutamatergic synaptic connectivity without notable changes to cell surface glutamate receptor. However, they based these data on spontaneous activity and local application of agonist. The classical approach would be to analyze the properties of miniature EPSC. Furthermore, they asses the amount of cell surface receptor to the amplitude of the current evoked by a local application of AMPA of NMDA on very few cells. This is not satisfactory.

We agree about the limitations of our current density experiments. It is for this reason that we did not put too much emphasis on these conclusions solely based on local application of agonists. This figure was included as more of a proof-of-concept of the presence of the specific currents. We have now re-phrased the conclusions in the revised version of the manuscript focussing solely on the qualitative aspects.

As the reviewer correctly points out, synaptic events and their parameters are more informative and, in this regard, amplitudes and kinetics of spontaneous EPSCs can equally be informative for changes to receptor expression/ clustering (Bellingham et al, J Physiol. 1998 Sep 15;511 (Pt 3)(Pt 3):861-9; Groc et al, J Neurosci. 2002 Jul 1;22(13):5552-62). Since our neurons do not receive high frequency EPSC inputs, and we do not observe changes to the amplitudes of spontaneous EPSCs (Figure 2D,I), AMPA receptor expression and clustering is mostly unaffected in these cells. This has now been addressed in the revised version of the manuscript.

7. Line 801-802 in figure 4, there is a mismatch between the lettering of the panels on the figure and the lettering in the legend. In panel F, how many waveforms have been averaged?

Thanks for spotting this. This has now been corrected and the number of waveforms averaged added to the figure panel in parentheses.

8. Line 218-219. The sentence regarding the conclusion of TEA sensitivity is not clear.

Authors show that there is less K⁺ A current but also show higher inhibition by TEA. Their experiments suggest a switch in K⁺ current component.

This has now been explained in greater detail in the revised version of the manuscript (see blue text) hopefully this is now clearer – essentially TEA reveals a disparity in the expression levels of specific K⁺ channels between D21 and T21

9. Immunolabeling section. Authors should include the dilution of antibodies.

Dilution of antibodies have now been added to the methods section of the manuscript.

10. Line 520-521, please provide the concentration of cAMP, GDNF and BDNF.

Concentrations of cAMP, GDNF and BDNF have now been added to the methods section of the manuscript as requested.

Reviewer #3 (Remarks to the Author)

In this manuscript the authors characterize intrinsic and network properties of neuronal cultures derived from iPSCs with normal chromosomal complement and with trisomy 21. Compared to controls glutamatergic synaptic activity is reduced (particularly AMPA mediated currents). Neurons with trisomy 21 show reduced excitability and deficits in A-current signaling

Overall assessment

This is a carefully conducted study addressing an important and timely issue in understanding Down Syndrome pathology. With increasing longevity of Down Syndrome patients, a better understanding of the neurological and cognitive aspects becomes ever more urgent and identification of potential drug targets more pressing. As detailed in the questions and comments, the usefulness in this approach heavily depends on the faithfulness with which the in-vitro approach recapitulates early development in-vivo. It would be useful for authors to expand on this a bit more in the manuscript's discussion. The lack of GABAergic activity should ideally also be discussed a bit more in detail – given that this is one of the more widely reported areas of functional defect in Down Syndrome patients.

We thank the reviewer for the positive assessment and finding that our work was “carefully conducted” and is “timely”; as suggested we have now elaborated on the faithfulness of our approach to early neurodevelopment, and the implications of a lack of GABA inhibition following our iPSC differentiations.

Main questions and comments :

1. Can the authors expand on the faithfulness with which iPSC derived neurons recapitulate embryonic development? Especially the absence of GABAergic neurons from most of their cultures seems intriguing, since other attempts in this direction observed inhibitory GABAergic activity.

This is an excellent point. As a model system, iPSCs are widely recognised to recapitulate many elements of early neurodevelopment (Shi et al, Nat Rev Drug Discov. 2017 Feb;16(2):115-130). This includes sophisticated changes involving specific brain area-like tissues. These cells have also provided us with enormous amounts of information on early human brain development in the context of disease (Brennand et al Nature. 2011 May

12;473(7346):221-5; Kawatani et al Commun Biol 4, 730 (2021)). While there are limitations to any approach using cellular models, our comparison between isogenic disomic and trisomic neurons should recapitulate early developmental changes that are captured by iPSCs as many groups have noted previously (eg - Weick et al, Proc. Natl. Acad. Sci. U.S.A. 110 (24) 9962-996, 2013; Kawatani et al, Commun Biol 4, 730, 2021).

We have now included quantification of glutamatergic neurons (Fig S1) revealing that over 93 % of our neurons are indeed glutamatergic which confirms the lack of GABAergic synaptic activity. This has several effects - it makes the cellular system less complex and has the reductionist advantage of studying a pure population of neurons; and in addition, it closely reflects those stages of brain development where GABA interneuron numbers are low in number (eg - before GABA interneuronal migration).

Importantly, our conclusions derived from using iPSCs are backed by our results from postmortem tissues and mathematical modelling which reinforce the key findings of our study giving us confidence of the validity of our results. These have now been discussed in the revised manuscript.

2. Line 566 - Whole cell electrophysiology of iPSC-derived neurons was carried out partially blinded of the genotype (two out of seven differentiations included). Can the authors clarify this statement? How many differentiations were blinded? Were the unblinded investigations done first?

All experiments (including two MEA runs) using cells from two differentiations were carried out blinded to the genotype. However, due to inherent difficulty associated with electrophysiology that make successful recording electrical activity of the highest quality a challenging process, we routinely carried out unblinded experiments and often confirmed outcomes with blinded experiments, as in this case. This has now been clarified in the manuscript.

3. Given that human transcriptomic data is available, what was the reason for the authors to not extend the analysis to a genome-wide comparison and to a comparison between iPSC derived neurons and neurons obtained from human cortex?

This is another excellent point. This is a direction we are pursuing but is beyond the scope of the current study. To reinforce our findings, we have now carried out additional experiments to demonstrate that the levels of expression for Kv4.3 are reduced, both at the mRNA and protein levels in our iPSC-derived neurons. These results and aspects of further human omics analyses are now discussed in the manuscript.

Detailed questions and comments:

1. "...and also because early-stage iPSC-derived neurons and immature neurons display developmentally regulated slow Ca²⁺ oscillations. Thus, MEA recordings ensured that only action potential-dependent changes to excitability were analysed". While MEA recordings might have been a sensible choice here, why is their inability to detect slow calcium oscillations a methodological advantage?

Early plasticity networks frequently exhibit spiking, and MEA allows us to measure spiking and assess spiking network synchrony directly. Given the abundance of slow Ca²⁺ oscillations at early ages, using Ca²⁺ imaging as a proxy for cellular spiking can be difficult. Therefore, a direct measure of spiking is important for establishing the developmental properties of the network. This has now been clarified in the manuscript.

In addition, we wanted to examine spike activity without any other complications, such as those presented by Ca²⁺ oscillations activating other signalling pathways, so the absence of such oscillations was considered to be advantageous for analysing spiking. This point is now clarified in the revised text,

2. Line 136. "As expected, in the absence of Mg²⁺, neurons showed greater synaptic activity but 10% of trisomic cells (compared to 0% disomic neurons) did not exhibit synaptic currents despite undergoing 0 Mg²⁺-induced membrane bursting (Figure 2F) thereby identifying a proportion of cells that fail to integrate into cortical networks during development." These findings would indicate an abundance of silent (NMDA only synapses). Would this also indicate a deficit in synaptic plasticity? Could such a deficit be causal for the reduced synaptic integration?

Yes, this is also a possibility and a good point. Although there is an absence of NMDA mediated EPSCs this could reflect a limitation on synaptic integration and/or limited glutamate release. Interestingly such cells show burst firing possibly propelled by NMDA receptors causing membrane depolarisation. As noted by the reviewer, this lack of active synapses could be important for diminishing excitatory NMDA-dependent synaptic plasticity with consequences for learning and memory. This is considered in the revised text.

3. What is the reason the cumulative data for Spontaneous activity in Fig2 is shown as violin plots and not standard cumulative histograms? These are easier to quantitatively compare and analyze.

Cumulative histograms have now been added to Figure S4.

Reviewer #4 (Remarks to the Author):

In the manuscript "Synaptic and intrinsic membrane defects disrupt early neural network dynamics in Down syndrome", Hannan et al used isogenic pairs of Down syndrome (DS) iPSC-derived neurons to identify early neurodevelopmental deficits. Using multielectrode array (MEA) and patch-clamp electrophysiology, the authors show that trisomy 21 (T21) neurons have reduced glutamatergic synaptic connectivity, altered Na⁺ and K⁺ channel function, impaired spike firing, and disrupted network synchrony. The authors further conclude downregulation of Kv4.3 in T21 excitatory neurons could be the underlying mechanism.

This work is important to the field. However, the manuscript presented in the current form needs to be substantially improved to support the conclusions.

We thank the reviewer for appreciating the importance of our work to the field. All concerns raised have now been addressed in the revised version of the manuscript which includes new data that supports the conclusions of our study. The point-by-point replies to the comments are below.

Major concerns:

1. The data characterizing neurons derived from both Trisomic 21 (T21) and isogenic Disomic 21 (D21) iPSCs should be included. The authors show neural stem cell (NSC) markers (e.g., SOX2, Nestin, PAX6), general neuronal markers (e.g., TUBB3, MAP2), and synapse marker staining. However, to show that their differentiated neuronal differentiation cultures give rise predominantly to glutamatergic neurons, the authors should also include

immunostaining for glutamatergic markers (e.g., VGLUT1 and VGLUT2 for excitatory neurons) as well as inhibitory neuron markers (e.g., GABA or GAD65/67).

This is a good point. We have now carried out staining of VGLUT1 in our neurons as they belong predominantly to the cortical subtype (Wojcik et al, Proc. Natl. Acad. Sci. U.S.A. 101 (18) 7158-7163, 2004). This reveals that over 93% of neurons are indeed excitatory. This together with our previous observations of the paucity of GABAergic synaptic activity in our preparations suggests that our neurons are indeed predominantly excitatory. These results have now been incorporated in the revised version of the manuscript.

2. The data to support the mechanism on KCND3 (Kv4.3) downregulation in T21 neurons is insufficient. KCND3 mRNA expression data shown in Figure 7 is obtained from post-mortem brain datasets. What is the expression profile of KCND3 and associated genes in the iPSC-derived neurons reported in this work? Without direct confirmation in these iPSC-derived neurons, it is difficult to draw conclusion. Additional verification should also be provided to show that the decreased mRNA level of KCND3 leads to reduced protein expression and impaired K⁺ channel membrane trafficking. Hence, additional experiments the authors should consider include (i) RNA-seq or qPCR in their iPSC-derived neurons to show that KCND3 mRNA is indeed decreased. (ii) Western blot and/or immunofluorescence to show Kv4.3 protein expression in T21 iPSC-derived neurons.

Thanks for raising this point. We have carried out additional experiments to measure the levels of KCND3 mRNA and Kv4.3 protein in our iPSC-derived neurons using qPCR and Western blot and immunofluorescence analyses. Consistent with our previous observations in postmortem tissues, the levels of KCND3 mRNA and Kv4.3 proteins are indeed reduced in our iPSC-derived neurons. Reassuringly, we do not find changes to Kv4.2 which confirms our post-mortem analysis and eliminates this isoform of TEA-sensitive K channels as being responsible for the deficit in A-currents. These results have now incorporated in the revised version of the manuscript.

3. Functional rescue data on Kv4.3 is lacking. For instance, would overexpression of KCND3 rescue the Kv4.3 phenotype in T21 neurons? Would a relatively selective Kv4.3 opener rescue the phenotype observed in T21 neurons?

We decided not to extend our studies to include rescue data for several reasons. Over-expression of K⁺ channels can have overt phenotypes such as neurotoxicity (Urrutia et al, Front Cell Neurosci. 2024 May 17;18:1406709) and neurodevelopmental conditions such as autism (Parcia-Junco-Clemente et al, Proc. Natl. Acad. Sci. U.S.A. 110 (45) 18297-18302) amongst others. Moreover, we are also unaware of any selective enhancers of Kv4.3 function. This has now been discussed in the manuscript.

4. In the discussion, the authors should acknowledge that their work is based on a simplified in vitro culture system composed almost exclusively of glutamatergic neurons (upon showing immunofluorescence staining of VGLUT1, etc, as suggested earlier). This is important as the simplified system is different from co-culture or in vivo animal models, where both inhibitory and excitatory neurons are present, among other cell types.

A good point, there are limitations to our iPSC approach which have been previously published. Importantly, our conclusions derived from iPSCs are supported by our results from postmortem tissues and mathematical modelling which reinforce the key findings of our study providing confidence in the validity of our results. Nevertheless, the limitations of our approach have now been discussed in the revised manuscript.

Minor points:

1. In Figure 3, Panel B, inset is only included for D21, but not for T21. Please add inset for T21.

Inset has now added to Fig 3B

2. In the Figure 3 legend, Line 801-802 seems to describe panels N, O, P, Q, which are not shown in Figure 3. Please double check.

Appreciate this – now corrected

Reply to Ref 4

The revised manuscript has addressed all of my concerns, except for the issue regarding the functional rescue data on Kv4.3. The rationale provided for not performing a rescue experiment is unconvincing, and the discussion of this point in the main text remains insufficient. Therefore, I would still recommend including functional rescue data in this manuscript.

We appreciate the inclination of the ref, but we have not included functional rescue experiments for the reasons we stated previously: a lack of a Kv4.3 opener; and the effect of overexpression of an ion channel gene on the phenotype of the cell making the interpretation of the functional outcome quite difficult.

Although less obvious, for potassium channel genes the overexpression driven phenotype can elicit aberrant/mutant phenotypes, such as neurotoxicity, further complicating the interpretation. Phenotypes caused by overexpression are not infrequent (Prelich 2012; *Genetics* 190(3):841–854. doi: [10.1534/genetics.111.136911](https://doi.org/10.1534/genetics.111.136911)) For these reasons we see an overexpression experiment as yielding potentially confusing outcomes. These points have been discussed further in the manuscript.